# Evidence for functional pre-coupled complexes of receptor heteromers and adenylyl cyclase

Gemma Navarro[1,2], Arnau Cordomí [3], Verónica Casadó-Anguera[2], Estefanía Moreno[2], Ning-Sheng Cai[4], Antoni Cortés[2], Enric I. Canela [2], Carmen W. Dessauer[5], Vicent Casadó[2], Leonardo Pardo [3], Carme Lluís[2] & Sergi Ferré[4]

G protein-coupled receptors (GPCRs), G proteins and adenylyl cyclase (AC) comprise one of the most studied transmembrane cell signaling pathways. However, it is unknown whether the ligand-dependent interactions between these signaling molecules are based on random collisions or the rearrangement of pre-coupled elements in a macromolecular complex. Furthermore, it remains controversial whether a GPCR homodimer coupled to a single heterotrimeric G protein constitutes a common functional unit. Using a peptide-based approach, we here report evidence for the existence of functional pre-coupled complexes of heteromers of adenosine $A_{2A}$ receptor and dopamine $D_2$ receptor homodimers coupled to their cognate Gs and Gi proteins and to subtype 5 AC. We also demonstrate that this macromolecular complex provides the necessary frame for the canonical Gs-Gi interactions at the AC level, sustaining the ability of a Gi-coupled GPCR to counteract AC activation mediated by a Gs-coupled GPCR.

[1] Department of Biochemistry and Physiology of the Faculty of Pharmacy of the University of Barcelona, 08028 Barcelona, Spain. [2] Department of Biochemistry and Molecular Biomedicine of the Faculty of Biology and Institute of Biomedicine of the University of Barcelona and Center for Biomedical Research in Neurodegenerative Diseases Network, 08028 Barcelona, Spain. [3] Laboratory of Computational Medicine, School of Medicine, Autonomous University of Barcelona, 08193 Bellaterra, Spain. [4] Integrative Neurobiology Section, National Institute on Drug Abuse, National Institutes of Health, Baltimore, MD 21224, USA. [5] Department of Integrative Biology and Pharmacology, McGovern Medical School, University of Texas Health Science Center, Houston, TX 77030, USA. These authors contributed equally: Gemma Navarro, Arnau Cordomí. Correspondence and requests for materials should be addressed to S.Fé. (email: sferre@intra.nida.nih.gov)

nteractions between G protein-coupled receptors (GPCRs), Gα and Gβγ protein subunits and adenylyl cyclase (AC) have been classically analyzed in the frame of 'collision-coupling' mechanisms, which implies they are freely mobile molecules in the plasma membrane able to couple by random collision. Binding of an agonist to its GPCR induces the binding and subsequent activation of the heterotrimeric G protein, which leads to the dissociation of Gα and Gβγ subunits and binding of free Gα subunit to AC, leading to its regulation[1]. However, accumulating experimental evidence suggests that GPCR activation commonly occurs without dissociation of the receptor from its G protein, without G-protein subunit dissociation and even with pre-coupling of the heterotrimeric G protein to AC (reviewed in ref.[2]). Moreover, growing evidence suggests that the pentameric complex formed by one GPCR homodimer (two identical protomers) and one heterotrimeric G protein constitutes a common GPCR functional unit[3–6]. Therefore, classical GPCR physiology needs to be revisited in the frame of pre-coupling mechanisms and GPCR oligomerization.

The topology of mammalian transmembrane AC consists of a variable cytoplasmic N terminus (NT) and two large cytoplasmic domains, C1 and C2, separated by two membrane-spanning domains, M1 and M2, each comprising six putative transmembrane domains (TMs)[7]. C1 and C2 interact to form the enzyme catalytic core at their interface and their arrangement allows, at least in theory, the simultaneous binding of their external sides to Gsα and Giα[8], providing the structural framework for the canonical antagonistic interaction between Gs-coupled and Gi-coupled receptors at the AC level of specific AC isoforms, including AC1, AC5, and AC6[2,7]. Gsα subunit binds to C2 and increases the affinity of C1 and C2, promoting catalysis, while Giα, by binding to C1, works in the opposite direction and counteracts AC activation[7].

It is becoming accepted that GPCRs can form heteromers[6,9], defined as macromolecular complexes composed of at least two different protomers with biochemical properties that are demonstrably different from those of its individual compoments[6]. Considering homodimers as main functional GPCR units, heteromers could be viewed as constituted by different interacting homodimers[6]. Of special functional significance could be those heteromers constituted by one homodimer coupled to a Gs/olf (Gs for short) protein and another different homodimer coupled to a Gi/o (Gi for short) protein. Our hypothesis is that such a "GPCR heterotetramer" would be part of a pre-coupled macromolecular complex that also includes AC, a necessary frame for the canonical antagonistic interaction at the AC level. Recent studies have provided experimental evidence for the existence of GPCR heterotetramers that fulfill this scheme, like the adenosine $A_{2A}$-dopamine $D_2$ receptor ($A_{2A}R$–$D_2R$) heterotetramer[10]. In the present study, using interfering peptides with amino acid sequences of TMs of adenosine $A_{2A}R$ and $D_2R$ and putative TMs of AC5, we provide evidence for the existence of functional pre-coupled complexes of $A_{2A}R$ and $D_2R$ homodimers, their cognate Gs and Gi proteins and AC5, and demonstrate that this macromolecular complex provides the sufficient but necessary condition for the canonical Gs–Gi interactions at the AC level.

## Results

### Symmetrical TM interfaces in the $A_{2A}R$-$D_2R$ heterotetramer.

To identify the arrangement of $A_{2A}R$ and $D_2R$ protomers in the heterotetramer (TMs involved in the homo and heterodimerization interfaces), we used synthetic peptides with the amino acid sequence of TMs 1–7 of $A_{2A}R$ and $D_2R$ (TMs and TM peptides are abbreviated TM 1, TM 2, … and TM1, TM2, … respectively) fused to the HIV transactivator of transcription (TAT) peptide,

which determines the orientation of the peptide when inserted in the plasma membrane (see ref.[11] and Methods section). Peptides were first tested in bimolecular fluorescence complementation (BiFC) experiments, in HEK-293T cells expressing receptors fused to two complementary halves of YFP (Venus variant; cYFP and nYFP). Functionality of all fused receptors was shown with cAMP accumulation experiments (Supplementary Fig. 1). Fluorescence was detected when cells were transfected with $A_{2A}R$-nYFP and $A_{2A}R$-cYFP cDNA (broken lines in Fig. 1a) or with $D_2R$-nYFP and $D_2R$-cYFP cDNA (broken lines in Fig. 1b), indicating the formation of both $A_{2A}R$-$A_{2A}R$ and $D_2R$-$D_2R$ homodimers. Notably, when BiFC assay was performed in the presence of TM peptides (Fig. 1a, b), fluorescence complementation of $A_{2A}R$-nYFP and $A_{2A}R$-cYFP was only significantly reduced in the presence of TM6 of $A_{2A}R$ (Fig. 1a; see Methods and Supplementary Fig. 2 for justification of the optimal concentration and time of incubation of the TM peptides). Similarly, only TM6 of $D_2R$ reduced fluorescence complementation of $D_2R$-nYFP and $D_2R$-cYFP (Fig. 1b). These results indicate that TM 6 forms part of a symmetric interface for both $A_{2A}R$ and $D_2R$ homodimers when expressed alone. The same results were obtained in cells expressing $A_{2A}R$-nYFP and $A_{2A}R$-cYFP co-transfected with non-fused $D_2R$ cDNA (Fig. 1a) or in cells expressing $D_2R$-nYFP and $D_2R$-cYFP co-transfected with non-fused $A_{2A}R$ cDNA (Fig. 1b). These results therefore indicate that TM 6 also forms part of a symmetric interface for both $A_{2A}R$ and $D_2R$ homodimers in the heterotetramer. Fluorescence was also detected in cells expressing $A_{2A}R$-nYFP and $D_2R$-cYFP (broken lines in Fig. 1c), indicating the formation of $A_{2A}R$–$D_2R$ heteromers. This fluorescence was only significantly reduced in the presence of TM4 and TM5 of both $A_{2A}R$ and $D_2R$ (Fig. 1c), suggesting a TMs 4/5 interface for $A_{2A}R$ and $D_2R$ heterodimer in the heterotetramer. Additional evidence of heteromer formation via TMs 4/5 was obtained from proximity ligation assay (PLA). This technique permits the direct detection of molecular interactions between two proteins without the need of fusion proteins. $A_{2A}R$–$D_2R$ heteromers were observed as red punctate staining in HEK-293T cells expressing both $A_{2A}R$ and $D_2R$ (Supplementary Fig. 3a–c). Pretreatment of cells with TM4 and TM5 of $A_{2A}R$ and $D_2R$ but not with TM6 or TM7 (negative control), significantly decreased PLA staining (Supplementary Fig. 3d), decreasing the number of stained cells and red spots per stained cell (Fig. 2a), supporting TMs 4/5 as the interface of the $A_{2A}R$–$D_2R$ heteromer.

In HEK-293T cells expressing both receptors, the $A_{2A}R$ agonist CGS21680 (100 nM; minimal concentration with maximal effect) significantly increased basal cAMP and the $D_2R$ agonist quinpirole (1 µM; minimal concentration with maximal effect) decreased forskolin-induced cAMP (Fig. 2b). Pertussis toxin, by catalyzing ADP-ribosylation of the alpha-subunit of Gi, impeded $D_2R$-mediated Gi activation and thus the ability of quinpirole to inhibit forskolin-induced cAMP accumulation (Fig. 2b). Cholera toxin, by selectively catalyzing ADP-ribosylation of the alpha-subunit of Gs and leading to persistent AC stimulation, impeded an additional effect of CGS21680 but left unaltered the Gi-mediated quinpirole-induced inhibition of forskolin-induced cAMP accumulation (Fig. 2b). These results support the coupling of $A_{2A}R$ and $D_2R$ to their respective cognate Gs and Gi proteins in the $A_{2A}R$–$D_2R$ heterotetramer. We could then demonstrate that neither $A_{2A}R$ or $D_2R$ activation leads to rearrangements of the TM interfaces in the $A_{2A}R$–$D_2R$ heterotetramer, since, in the presence of CGS21680 (100 nM) or quinpirole (1 µM), fluorescence in cells expressing $A_{2A}R$-nYFP and $D_2R$-cYFP was still selectively reduced by TM4 and TM5 of $A_{2A}R$ and $D_2R$ (Fig. 1c). Similarly, $A_{2A}R$ activation by CGS21680 (Fig. 1a) or $D_2R$ activation by quinpirole (Fig. 1b) did not modify the corresponding specific homomer TM 6 interface determined in ligand-free experiments.

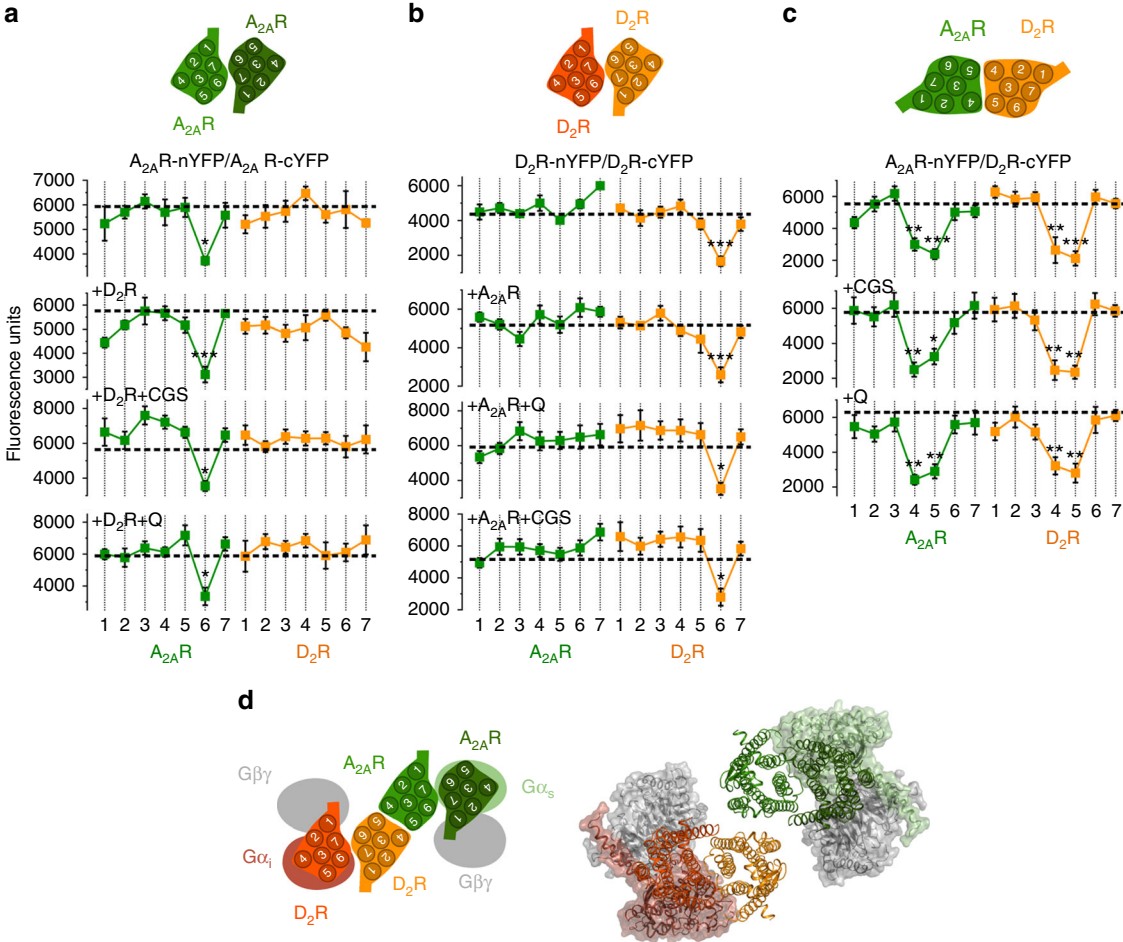

**Fig. 1** Quaternary structure of $A_{2A}R$-$D_2R$ heterotetramer coupled to Gs and Gi proteins. **a**–**c** BiFC experiments in HEK-293T cells transfected with $A_{2A}R$-nYFP (0.5 μg) and $A_{2A}R$-cYFP (0.5 μg) cDNA in the absence or presence of $D_2R$ cDNA (0.5 μg) (**a**), with $D_2R$-nYFP (0.75 μg) and $D_2R$-cYFP (0.75 μg) cDNA in the absence or the presence of $A_{2A}R$ cDNA (0.4 μg) (**b**) or with $A_{2A}R$-nYFP (0.6 μg) and $D_2R$-cYFP (0.6 μg) cDNA (**c**); cells were treated for 4 h with medium (broken lines) or 4 μM of indicated TM peptides (numbered 1–7) of $A_{2A}R$ (green squares) or $D_2R$ (orange squares) before addition of medium, CGS21680 (CGS; 100 nM) or quinpirole (Q; 1 μM); fluorescence was detected at 530 nm and values (in means ± SEM) are expressed as fluorescence arbitrary units ($n = 8$, with triplicates); *, **, and *** represent significantly lower values as compared to control values ($p < 0.05$, $p < 0.01$ and $p < 0.001$, respectively; one-way ANOVA followed by Dunnett's multiple comparison tests). **d** Computational model of the $A_{2A}R$-$D_2R$ heterotetramer built using the experimental interfaces predicted in panels (**a**–**c**) (TMs 4/5 for heterodimerization and TM 6 for homodimerization) with Gs and Gi binding to the external protomers; schematic slice-representation (left) and the constructed molecular model (right; with the same color code as the schematic slice-representation), viewed from the extracellular side

We then constructed a molecular model of the $A_{2A}R$–$D_2R$ heterotetramer (Fig. 1d), considering: (i) the crystal structures of GPCRs and G proteins, as well as homology models (see Methods section); (ii) the structural details of TM interfaces of GPCR oligomers, observed in crystal structures[12] as well as predicted by molecular dynamics simulations (see Methods section); (iii) the results from BiFC experiments with interfering TM peptides; (iv) the general assumption of a common minimal functional unit of GPCRs constituted by a homodimer coupled to its cognate G protein (see Introduction section); (v) the suggested tetrameric structure of the $A_{2A}R$–$D_2R$ heteromer constituted by two interacting homodimers, from previous results obtained with bioluminescence resonance energy transfer (BRET) experiments with complementation of both the donor and the acceptor biosensors[10]; and (vi) the previously enunciated assumption about the necessity of a simultaneous activation of Gs and Gi coupled to the interacting catalytic domains of the same molecule of AC for a canonical antagonistic interaction[8]. This resulted in one minimal computational solution that accommodates the TMs 4/5 interface for $A_{2A}R$–$D_2R$ heterodimerization and the TM 6

interface for both $A_{2A}R$–$A_{2A}R$ and $D_2R$–$D_2R$ homodimerization (see Methods and Supplementary Fig. 4). The existence of these interfaces implies two internal interacting $A_{2A}R$ and $D_2R$ protomers and two external $A_{2A}R$ and $D_2R$ protomers in which the α-subunits of Gi and Gs bind to the corresponding external protomers of the $D_2R$ or $A_{2A}R$ homodimers. This would be the only feasible configuration to avoid any steric clash between the two G proteins simultaneously bound to the complex. Finally, the model also predicts a large distance between both βγ-subunits (Fig. 1d).

**Asymmetrical TM interfaces of the heterotetramer with AC5.** Although several studies have provided direct evidence for pre-coupling between G protein subunits and AC[7,13–15], specifically with the AC NT[7,14], to our knowledge, the existence of pre-coupling between TMs of a GPCR and TMs of AC had not been previously addressed. We first analyzed the ability of AC5 to establish direct intermolecular interactions with $A_{2A}R$ or $D_2R$ or with $A_{2A}R$–$D_2R$ heteromers via saturation BRET experiments in

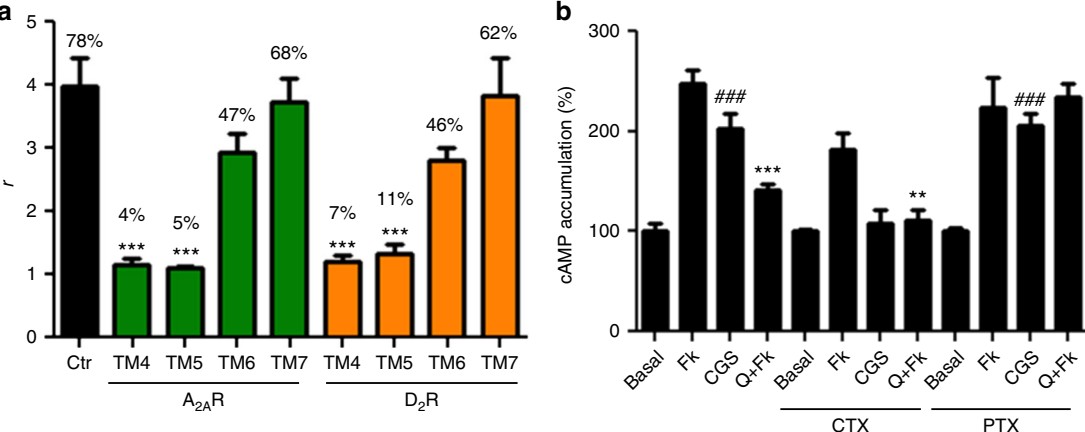

**Fig. 2** Functional $A_{2A}R$–$D_2R$ heterotetramers in transfected cells. **a** Quantification from PLA experiments (see Supplementary Fig. 1) performed in HEK-293T cells transfected with 0.4 μg of $A_{2A}R$ and 0.5 μg of $D_2R$ cDNA treated for 4 h with medium (control) or 4 μM of indicated TM peptides of $A_{2A}R$ or $D_2R$; values are expressed as the ratio between the number of red spots representing heteromers in confocal microscopy images and the number of cells showing spots (*r*) (30–50 cells from three independent preparations); % values represent the percentage of cells showing one or more red spots; ***$p < 0.001$, as compared to control (one-way ANOVA followed by Dunnett's multiple comparison tests). **b** cAMP production in HEK-293T cells transfected as in (**a**); cells were incubated overnight with vehicle or pertussis toxin (PTX; 10 ng/ml), or for 2 h with cholera toxin (CTX; 100 ng/ml), and exposed to CGS21680 (CGS; 100 nM) or quinpirole (Q; 1 μM) in the absence or in the presence of forskolin (Fk; 0.5 μM), respectively; values are expressed as percentage over cAMP accumulation in non-treated cells (basal) ($n = 5$–7, with triplicates); ###$p < 0.001$, as compared to basal values; ** and ***$p < 0.01$ and $p < 0.001$ as compared to Fk, respectively; one-way ANOVA followed by Tukey's multiple comparison tests. Results are always represented as means ± SEM

the absence of ligands (results are always shown as means ± SEM). Clear-cut saturation BRET curves were obtained with HEK-293T cells transfected with a constant amount of $A_{2A}R$ fused to *Renilla* Luciferase ($A_{2A}R$-RLuc) cDNA and increasing quantities of AC5 fused to YFP (AC5-YFP) cDNA (Fig. 3a; $BRET_{max} = 54 \pm 6$ mBU and $BRET_{50} = 42 \pm 13$) or with cells transfected with a constant amount of $D_2R$-RLuc cDNA and increasing amounts of AC5-YFP cDNA (Fig. 3b; $BRET_{max} = 38 \pm 5$ mBU and $BRET_{50} = 28 \pm 14$), indicating that AC5 interacts with $A_{2A}R$ or $D_2R$ in the absence of ligands. Also, saturation BRET curves were obtained when HEK-293T cells transfected with $A_{2A}R$-RLuc and increasing amounts of AC5-YFP cDNAs were co-transfected with $D_2R$ cDNA (Fig. 3c; $BRET_{max} = 39 \pm 3$ mBU and $BRET_{50} = 24 \pm 8$) or when cells transfected with $D_2R$-RLuc and increasing amount of AC5-YFP cDNAs were co-transfected with $A_{2A}R$ cDNA (Fig. 3d; $BRET_{max} = 30 \pm 2$ mBU and $BRET_{50} = 20 \pm 7$). All saturation BRET curves were best-fitted to a monophasic model. We also verified that over-expression of AC5 did not alter $A_{2A}R$–$D_2R$ heteromerization with BRET experiments in HEK-293T cells transfected with $A_{2A}R$-Rluc (0.4 μg) and $D_2R$-YFP (0.6 μg) and increasing amounts of AC5 cDNA. No BRET differences were observed between the results obtained with 0, 0.3, 1.0 and 3.0 μg of AC5 cDNA ($56 \pm 7$, $53 \pm 6$, $53 \pm 3$, and $52 \pm 4$ mBU, respectively). Altogether, these results suggest that AC5 oligomerize with $A_{2A}R$-$D_2R$ heteromers in the absence of ligands.

Next, we performed BiFC assays in HEK-293T cells expressing AC5-nYFP, $A_{2A}R$-cYFP, and $D_2R$ (Fig. 3e) as well as AC5-nYFP, $D_2R$-cYFP and $A_{2A}R$ (Fig. 3f). Normal functionality of AC5-YFP has been previously reported[16]. Significant fluorescence was detected in all cases, providing additional support to direct interactions between AC5 and $A_{2A}R$–$D_2R$ heteromers (broken lines in Fig. 3e, f). To determine the possible involvement of receptor TMs in the $A_{2A}R$–$D_2R$ heterotetramer-AC5 interface, we performed BiFC experiments with all different $A_{2A}R$ (Fig. 3e) or $D_2R$ (Fig. 3f) TM peptides. In the absence of ligands, pretreatment of cells with TM1, TM5, or TM6 of $A_{2A}R$ significantly decreased complementation between AC5 and $A_{2A}R$ (Fig. 3e, top panel). Similarly, pretreatment with TM1,

TM4, TM5, or TM6 of $D_2R$ significantly decreased complementation between AC5 and $D_2R$ (Fig. 3f, top panel). This suggests a discrete interaction between TM1 of both receptors with AC5. Since TMs 4–5 of the inner receptor protomers and TMs 6 of inner and outer receptor protomers participate in homo- and heterodimerization (see above), respectively, their apparent involvement in the interactions with AC5 must be indirect, implying that the optimal interaction of the $A_{2A}R$–$D_2R$ heterotetramer and AC5 requires the optimal quaternary structure of the heterotetramer. When BiFC experiments were performed in the presence of CGS21680 (100 nM, Fig. 3e, bottom panel) or quinpirole (1 μM, Fig. 3f, bottom panel), the pattern of interfering synthetic peptides changed: In addition to TM5 and TM6 of $A_{2A}R$ and $D_2R$, TM7 of $A_{2A}R$ and TM2 of $D_2R$ decreased fluorescence complementation in the presence of CGS21680 and quinpirole, respectively, while TM1 of $A_{2A}R$ and $D_2R$ were no longer effective (Fig. 3e, f).

We then investigated the involvement of TMs of AC5 TMs in the oligomerization with $A_{2A}R$–$D_2R$ heteromers. Since the structure of M1 and M2 domains of any AC isoform is unknown, we used five commonly used algorithms to predict their most probable TMs (Supplementary Table 1). All algorithms predicted the same six TMs for the M2 domain (TM 7 to TM 12), but there was discrepancy on the predicted TMs of the M1 domain. Taking into account the orientation of the predicted TM helices, only Uniprot and TOPCONS solutions were compatible with the well-established intracellular N-terminal and C-terminal domains of AC5[7]. First, TM peptides mimicking right-oriented TMs derived from Uniprot predictions (abbreviated TM1 to TM12) were tested for their ability to destabilize complementation in HEK-293T cells expressing AC5-nYFP, $A_{2A}R$-cYFP, and $D_2R$ (Fig. 4a), as well as AC5-nYFP, $D_2R$-cYFP and $A_{2A}R$ (Fig. 4b). In the absence of agonists, pretreatment of cells with TM1 or TM12 significantly decreased complementation between AC5 and $A_{2A}R$, while TM5 showed a small but not significant decrease (Fig. 4a, top panel). Similarly, pretreatment with TM6 or TM12 significantly decreased complementation between AC5 and $D_2R$ while TM5 again showed a small but not significant decrease (Fig. 4b, top panel). Remarkably, when BiFC

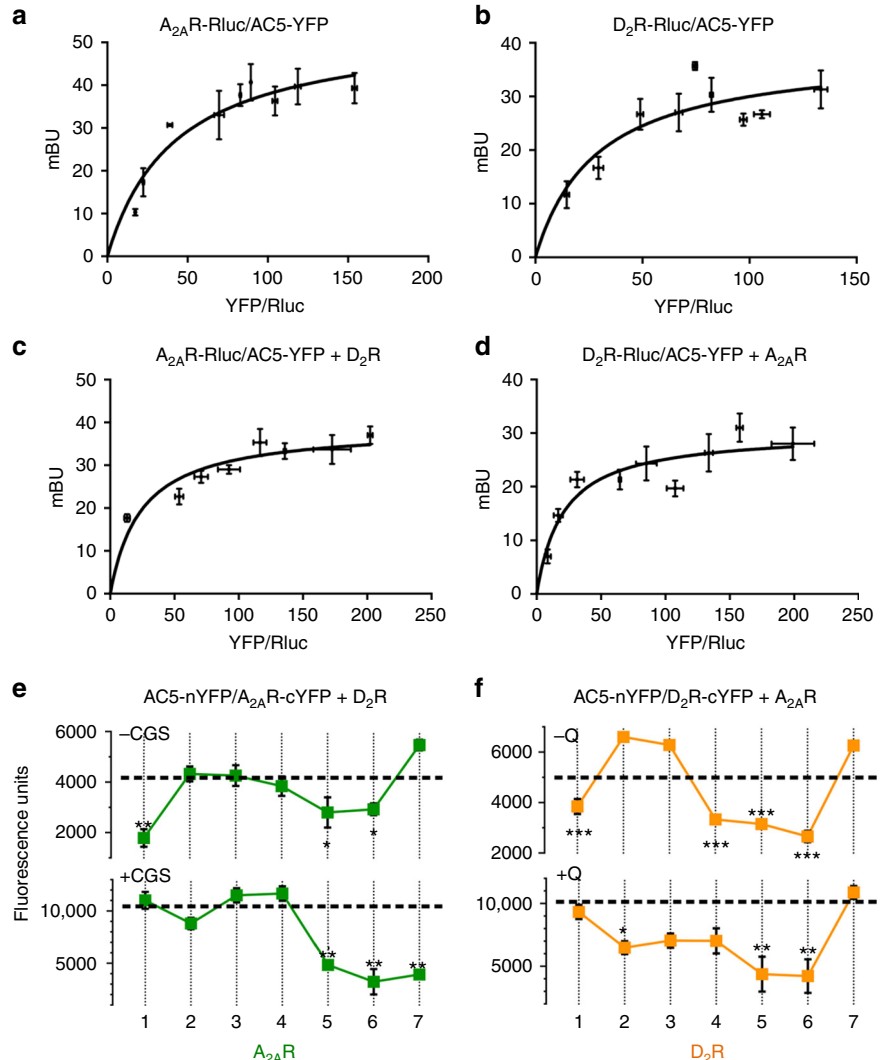

**Fig. 3** Involvement of receptor TMs in $A_{2A}R$–$D_2R$ heterotetramer-AC5 oligomerization. **a**–**d** BRET saturation experiments in HEK-293T cells transfected with 0.5 µg of $A_{2A}R$-Rluc cDNA and increasing amounts of AC5-YFP cDNA (0.3–2.5 µg) not co-transfected (**a**) or co-transfected (**c**) with $D_2R$ cDNA (0.5 µg), or with 0.75 µg of $D_2R$-Rluc cDNA and increasing amounts of AC5-YFP cDNA (0.3–2.5 µg) not co-transfected (**b**) or co-transfected (**d**) with $A_{2A}R$ cDNA (0.4 µg); the relative amount of BRET is given as a function of 1000× the ratio between the fluorescence of the acceptor (YFP) and the luciferase activity of the donor (Rluc) and expressed as milli BRET units (mBU) (6–8 experiments, with duplicates, grouped as a function of the amount of BRET acceptor). **e**, **f** BiFC experiments in HEK-293T cells transfected with AC5-nYFP (0.75 µg), $A_{2A}R$-cYFP (0.5 µg) and $D_2R$ (0.75 µg) cDNA (**e**) or AC5-nYFP (0.75 µg), $D_2R$-cYFP (0.75 µg) and $A_{2A}R$ (0.4 µg) cDNA (**f**); cells were treated for 4 h with medium (dotted lines) or 4 µM of indicated TM peptides (numbered 1–7) of $A_{2A}R$ (**e**) or $D_2R$ (**f**) before addition of medium, CGS21680 (CGS; 100 nM; **e**) or quinpirole (Q; 1 µM; **f**); fluorescence was detected at 530 nm and values are expressed as arbitrary fluorescent units ($n = 8$, with triplicates); *, ** and *** represent significantly lower values as compared to control values ($p < 0.05$, $p < 0.01$ and $p < 0.001$, respectively; one-way ANOVA followed by Dunnett's multiple comparison tests). Results are always represented as means ± SEM

experiments were performed in the presence of CGS21680 (100 nM, Fig. 4a, bottom panel) or quinpirole (1 µM, Fig. 4b, bottom panel), the pattern of interfering synthetic peptides dramatically changed. When receptors were activated, TM1, TM2, TM3, TM5 and TM6 significantly decreased fluorescence complementation between AC5-nYFP and $A_{2A}R$-cYFP and between AC5-nYFP and $D_2R$-cYFP. The results imply a major rearrangement of the membrane-spanning domains of the activated pre-coupled complex with an increase in the number of TMs of AC5 directly or indirectly involved in the oligomerization with the $A_{2A}R$–$D_2R$ heterotetramer.

Opposite-oriented TM peptides, abbreviated as TM2n, TM3n, TM4n, TM5n and TM6n, were tested to examine the specificity of their destabilizing effect (see Supplementary Table 2), which should insert in the membrane in the opposite direction and act

as scrambled control peptides. The peptides were tested in HEK-293T cells expressing AC5-nYFP, $A_{2A}R$-cYFP, and $D_2R$ (Fig. 4c) as well as AC5-nYFP, $D_2R$-cYFP, and $A_{2A}R$ (Fig. 4d) in the absence or in the presence of agonists. The same as TM4, TM4n did not have a significant effect, and TM2n, TM3n and TM6n did behave as negative controls to their opposite-oriented peptides, since they did not decrease AC5-nYFP-$A_{2A}R$-cYFP or AC5-nYFP-D2R-cYFP complementation in the absence (Fig. 4c, d, top panels) or in the presence (Fig. 4c, d, bottom panels) of agonists. Intriguingly, both TM5 and the opposite-oriented TM5n were able to decrease AC5-nYFP-$A_{2A}R$-cYFP and AC5-nYFP-D2R-cYFP complementation (Fig. 4c, d). Importantly, TM5 and TM5n had the lowest hydrophobicity as compared to all the other putative TM sequences (Supplementary Table 1), decreasing the probability of being embedded in the membrane bilayer[17]. This

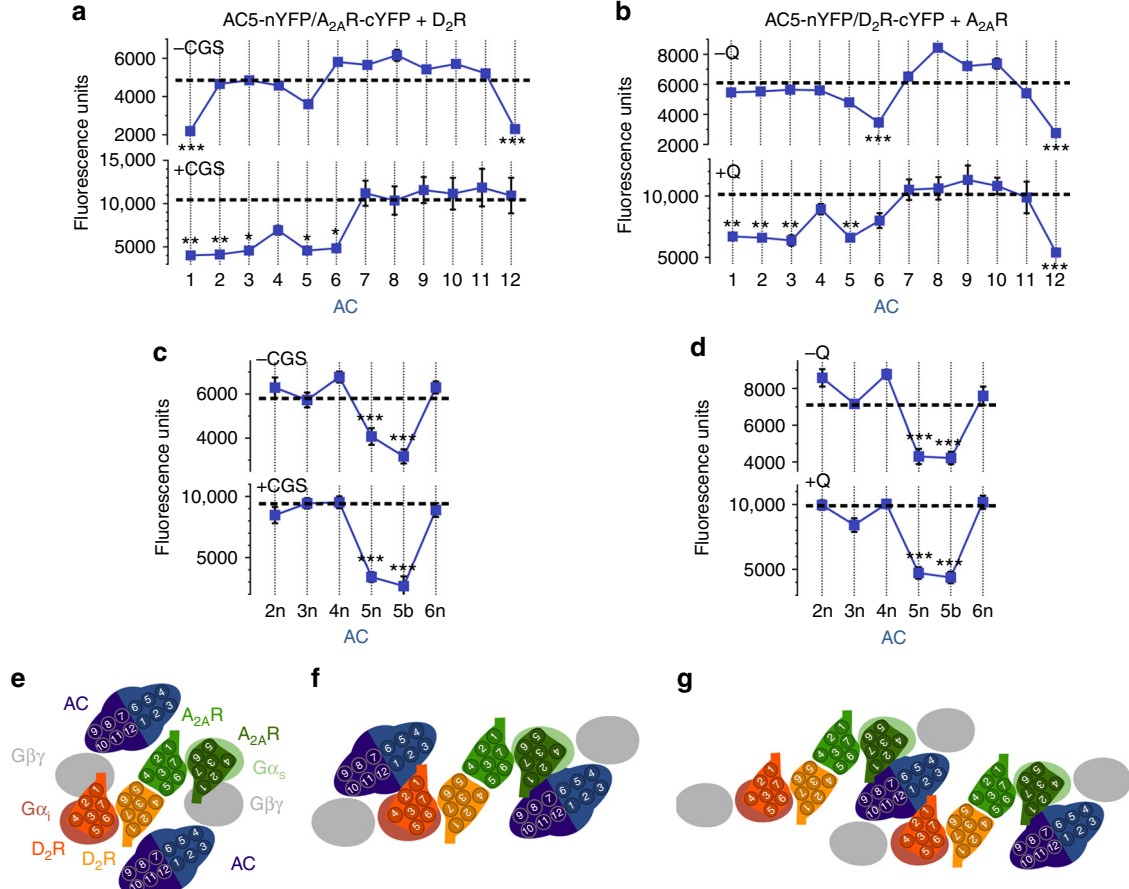

**Fig. 4** Involvement of AC5 TMs in $A_{2A}R$-$D_2R$ heterotetramer-AC5 oligomerization. **a–d** BiFC experiments in HEK-293T cells transfected with AC5-nYFP (0.75 μg), $A_{2A}R$-cYFP (0.5 μg) and $D_2R$ (0.75 μg) cDNA (**a**, **c**) or AC5-nYFP (0.75 μg), $D_2R$-cYFP (0.75 μg) and $A_{2A}R$ (0.4 μg) cDNA (**b**, **d**); cells were treated for 4 h with medium (dotted lines) or 4 μM of indicated TM peptides predicted from Uniprot algorithm (numbered 1–12) (**a**, **b**) or control peptides (numbered 2n−6n and 5b; see text) (**c**, **d**), before addition of medium, CGS21680 (CGS; 100 nM) or quinpirole (Q; 1 μM); fluorescence was detected at 530 nm and values (in means ± SEM) are expressed as arbitrary fluorescent units ($n = 8$, with triplicates); *, ** and *** represent significantly lower values as compared to control values ($p < 0.05$, $p < 0.01$, and $p < 0.001$, respectively; one-way ANOVA followed by Dunnett's multiple comparison tests). **e–g** Schematic slice-representations of $A_{2A}R$–$D_2R$ heterotetramer-AC5 models: heterotetramer coupled with two AC5 molecules in the absence (**e**) and in the presence (**f**) of agonists; extension of the agonist-bound complex with a second $A_{2A}R$–$D_2R$ heterotetramer, with simultaneous binding of both Gαs and Gαi to the central C1 and C2 domains of AC5 (**g**). Schematic slice-representation viewed from the extracellular side of the $A_{2A}R$–$D_2R$ heterotetramer in complex with Gs, Gi, and AC5 in the absence and presence of agonists are shown in Supplementary Fig. 6

could indicate that the AC5 325–345 amino acid sequence forms part of the second intracellular loop (IL2), which could establish direct or indirect intermolecular interactions with the $A_{2A}R$-$D_2R$ heteromer. Then, the 348–368 aa sequence predicted by the TOPCONS algorithm (TM 5b in Supplementary Table 1), which has the right orientation, becomes a very plausible TM that could interact with the $A_{2A}R$–$D_2R$ heterotetramer. In fact, TM5b peptide significantly decreased AC5-nYFP-$A_{2A}R$-cYFP or AC5-nYFP-D2R-cYFP complementation in the absence or in the presence of agonists (Fig. 4c,d). In agreement with this interpretation, a scrambled TM5-TM5n peptide (AC5-TM5s in Supplementary Table 2) did not decrease AC5-nYFP-$A_{2A}R$-cYFP or AC5-nYFP-D2R-cYFP complementation in the absence of ligands (93 ± 7, and 95 ± 6%, respectively, in means ± SEM and expressed as percentage of change of fluorescent values without peptide; $n = 9$, with triplicates). As additional controls, we also tested AC5 TM1 to TM12 peptides on A2AR-nYFP-D2R-cYFP complementation and all the D2R TM and A2AR TM peptides on AC5-nYFP-A2AR-cYFP and AC5-nYFP-D2R-cYFP complementation, respectively, in the absence of ligands; no changes in BiFC were observed under any condition

(Supplementary Fig. 5). Considering TM 1, TM 2, TM 3, TM 4, TM 5b, and TM 6 as the six putative TMs of the M1 domain of AC5, altogether these results indicate that TM 1 and TM 6, as well as IL2 and TM 5b, are involved in pre-coupling of $A_{2A}R$-$D_2R$ heterotetramer and AC5 in the absence of agonists. Upon $A_{2A}R$ or $D_2R$ activation there is a rearrangement with an apparent participation of almost all TMs of the M1 domain.

**Two $A_{2A}R$–$D_2R$ heterotetramers and two AC5 molecules**. It seems reasonable to hypothesize that the membrane-spanning domains of AC5 are formed by two interacting antiparallel six-helix-bundle domains (M1–M2) with an elliptical ring shape[7]. In the absence of ligands, since it is not feasible that TM 1 from both $A_{2A}R$ and $D_2R$ interact simultaneously with the same TM 5 or TM 12 or the same IL2 of a single AC5 molecule, this suggests the presence of two AC5 molecules simultaneously binding to the $A_{2A}R$–$D_2R$ heterotetramer in complex with Gi and Gs, possibly with TM1 of $D_2R$ and TM 1 of $A_{2A}R$ interacting specifically with TM 1 and TM 6 of AC5, respectively (Fig. 4e). The ability of peptides that mimic TM 5, TM 12, and IL2 of AC to destabilize

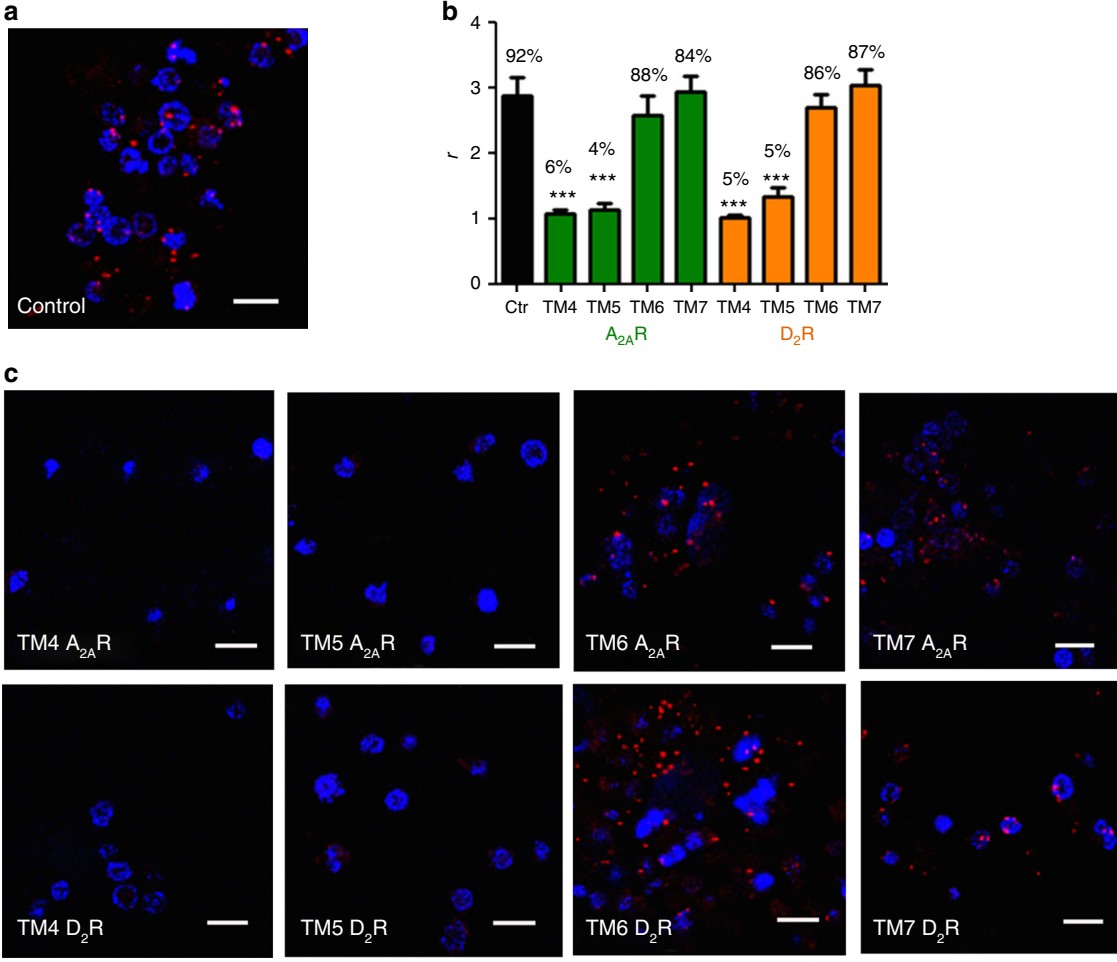

**Fig. 5** $A_{2A}R$-$D_2R$ heterotetramer expression in striatal neurons in culture. Proximity ligation assay (PLA) in rat striatal primary cultures. **a**, **c** Confocal microscopy images (superimposed sections) are shown in which $A_{2A}R$-$D_2R$ heteromers appear as red spots. Primary cultures were treated for 4 h with medium (**a**) or 4 μM of indicated TM peptides (numbered 1–7) of $A_{2A}R$ or $D_2R$ (**c**); cell nuclei were stained with DAPI (blue); scale bars: 20 μm. **b** Quantification from PLA experiments: values (in means ± SEM) are expressed as the ratio between the number of red spots and the number of cells showing spots ($r$) (20–30 neurons from three independent preparations); % values represent the percentage of cells showing one or more red spots; ***$p$ < 0.001, as compared to control (one-way ANOVA followed by Dunnett's multiple comparison tests)

oligomerization between AC5 and the $A_{2A}R$–$D_2R$ heterotetramer might depend on an indirect modification of their discrete asymmetrical interfaces.

It is well established that the $G\alpha$ binding site for $G\beta\gamma$ overlaps with the $G\alpha$ binding sites for the effector, the cytoplasmic domains C1 and C2 of AC. During G protein activation, $G\beta\gamma$ relative movement promotes $G\alpha$ binding to AC[18,19]. These swapping interactions can take place within the frame of the $A_{2A}R$–$D_2R$ heterotetramer with two AC5 molecules binding simultaneously to Gs and Gi in the complex (Fig. 4e, f). The rearrangement of TM interfaces between the $A_{2A}R$–$D_2R$ heterotetramer and AC5 upon receptor activation occurs simultaneously with the rearrangement of the $G\beta\gamma$ subunit, by its established stable coupling with the NT of AC5[16], which facilitates the interaction between the $G\alpha$ subunit and its corresponding catalytic AC5 domain. This rearrangement in the frame of the heteromer gives a computational molecular model of activated complex schematized in Fig. 4f. Details about the model are shown in Supplementary Fig. 6. However, within the frame of the constraints imposed by a pre-coupled $A_{2A}R$–$D_2R$ heterotetramer-Gs-Gi-AC5 complex, a single $A_{2A}R$–$D_2R$ heterotetramer cannot accomplish the model proposed by Dessauer et al.[8], in which one Gs and one Gi bind simultaneously to one single AC5 (see below). Therefore, we propose that AC5 should oligomerize

with an additional $A_{2A}R$–$D_2R$ heterotetramer (Fig. 4g). The results with interfering peptides, together with the proposed simultaneous binding of Gs and Gi to AC5, suggest a minimal functional complex composed of two $A_{2A}R$–$D_2R$ heterotetramers and two AC5 molecules (Fig. 4g).

**The canonical Gs–Gi antagonistic interaction.** To corroborate the proposed model we studied the functional characteristics of the $A_{2A}R$–$D_2R$ heterotetramer-AC5 complex in rat striatal neuronal primary cultures, which express endogenous $A_{2A}R$–$D_2R$ heteromer complexes[20]. Furthermore, AC5 is the predominant AC subtype in striatal neurons[21]. First, we analyzed by PLA the expression of $A_{2A}R$–$D_2R$ heteromers, as well as the ability of the synthetic peptides mimicking the TMs of $A_{2A}R$ and $D_2R$ to modify the quaternary structure of the endogenous $A_{2A}R$–$D_2R$ heterotetramer. $A_{2A}R$–$D_2R$ heteromers were observed as red punctate staining in neuronal cells (Fig. 5a, b). As expected, pretreatment of cells with TM4 and TM5 of $A_{2A}R$ and $D_2R$, but not with TM6 or with TM7, produced a significant decrease in the number of red spots per cell (Fig. 5b, c). These results mirrored those obtained in HEK-293T cells (see Fig. 2a and Supplementary Fig. 3), confirming the same TMs 4/5 interface of

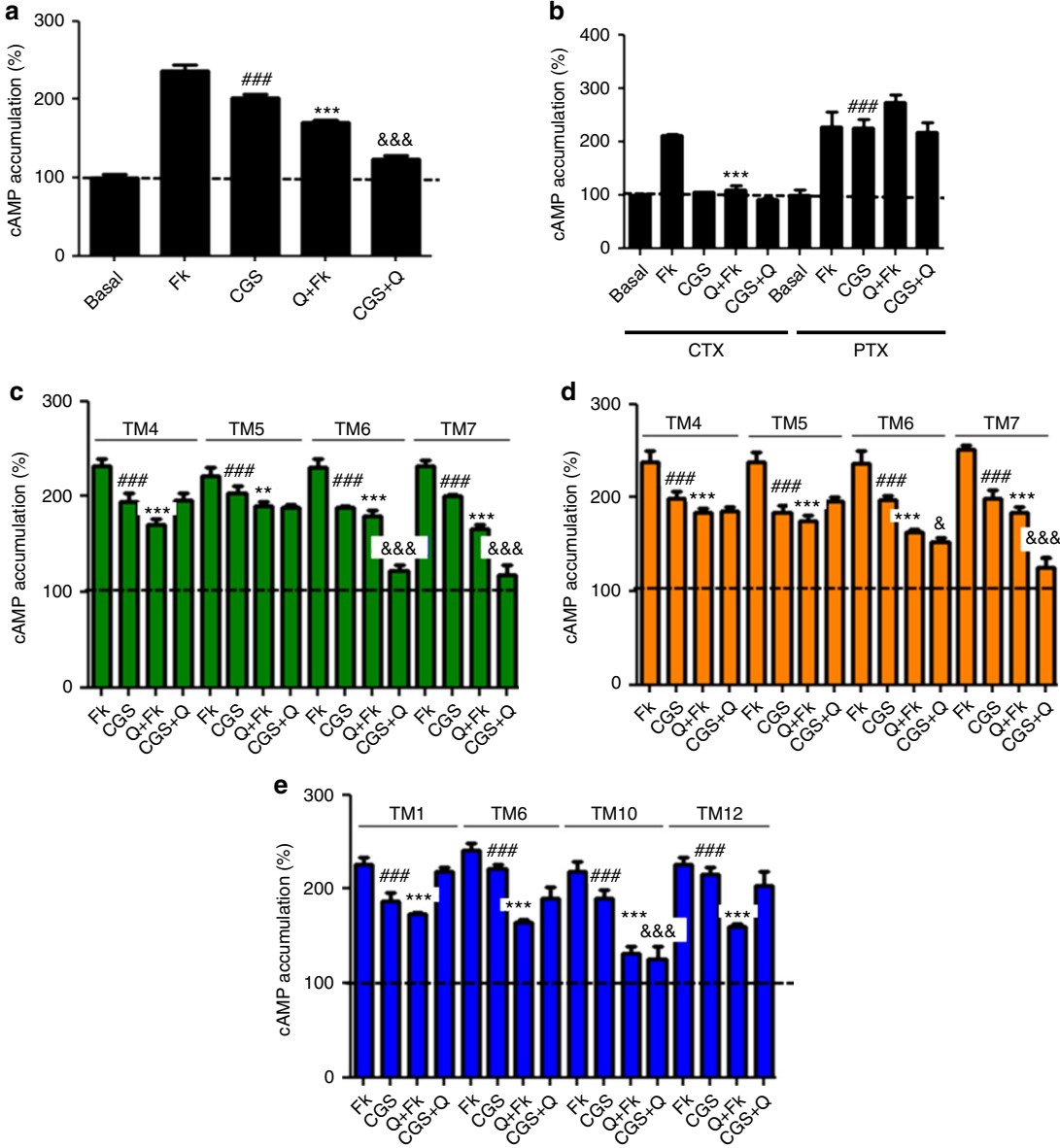

**Fig. 6** Canonical Gs–Gi antagonistic interaction in striatal neurons in culture. **a**, **b** cAMP production determined in rat striatal primary cultures incubated overnight with vehicle (**a**) or with pertussis toxin (PTX; 10 ng/ml), or for 2 h with cholera toxin (CTX; 100 ng/ml) (**b**), and exposed to CGS21680 (CGS; 100 nM), quinpirole (Q; 1 μM) or both in the absence or in the presence of forskolin (Fk; 0.5 μM), respectively. **c**–**e** cAMP production determined in rat striatal primary cultures incubated 4 h with 4 μM of indicated TM peptides of $A_{2A}R$ (**c**), $D_2R$ (**d**), or AC5 (**e**) and exposed to agonists as in **a**, **b**. Values (in means ± SEM) are expressed as percentage of cAMP accumulation in non-treated cells (basal) (n = 5–7, with triplicates); [###] $p < 0.001$, as compared to basal values; ** and ***$p < 0.01$ and $p < 0.001$ as compared to Fk, respectively; [&], [&&&] $p < 0.05$ and $p < 0.001$ as compared to CGS, respectively; one-way ANOVA followed by Tukey's multiple comparison tests

$A_{2A}R$–$D_2R$ heteromers in striatal cells and that TM6 does not destabilize heterodimerization. PLA experiments were also performed with a recently characterized AC5 antibody[22]. $A_{2A}R$-AC5 and $D_2R$-AC5 complexes could be revealed as red punctate staining in neuronal cells (Supplementary Fig. 7). Next, we measured cAMP production to analyze the functional characteristics of the $A_{2A}R$–$D_2R$ heteromer and the effect of the interfering peptides. As expected, CGS21680 (100 nM) increased the synthesis of cAMP (Fig. 6a) and quinpirole (1 μM) decreased forskolin-induced cAMP accumulation (Fig. 6a). Pertussis toxin, selectively counteracted the ability of quinpirole to inhibit forskolin-induced cAMP (Fig. 6b), while cholera toxin impeded the activating effect of CGS21680 while leaving unaltered quinpirole-induced inhibition of forskolin-induced AC5

activation (Fig. 6b). Simultaneous exposure to both agonists demonstrated the ability of quinpirole to inhibit the effect of CGS21680, revealing the canonical Gs–Gi interaction at AC5 (Fig. 6a).

Accumulation of cAMP was also determined in striatal cultures upon exposure to ligands and interfering TM peptides. Pretreatment with TM7 (as negative control) or with TM6 of $A_{2A}R$ or $D_2R$ did not modify receptor signaling or the canonical interaction (Fig. 6c, d). In contrast, although pretreatment with TM4 and TM5 of $A_{2A}R$ (Fig. 6c) or $D_2R$ (Fig. 6d) did not modify receptor signaling, it blocked the canonical interaction (Fig. 6c, d). These results indicate that TMs that destabilize receptor heteromerization do not disrupt the individual functional interactions between the receptors and AC5, most probably

because of stable pre-coupling between the G proteins and AC5, as recently demonstrated for the specific $G\alpha_{olf}\beta_2\gamma_7$-AC5 complex in the rodent striatum[22]. Nevertheless, the peptides that destabilize receptor heteromerization alter the correct coupling of AC5 to the complex that allows the simultaneous binding of $G\alpha s$ and $G\alpha i$ subunit to AC, impeding the canonical interaction. In conclusion, $A_{2A}R$–$D_2R$ heteromerization is a necessary condition for the canonical antagonistic interaction between Gs-coupled $A_{2A}R$ and Gi-coupled $D_2R$ at AC in striatal neurons in culture. In agreement with this conclusion, cAMP accumulation induced by CGS21680 was not counteracted by an agonist of dopamine $D_4R$, which does not heteromerize with $A_{2A}R$[23] (Supplementary Fig. 8). Finally, pretreatment of striatal cultures with interfering peptides TM1, TM6 or TM12 of AC5 did not change receptor signaling but also blocked the canonical interaction, while TM10 was ineffective (Fig. 6e). These results confirm the involvement of the AC5 subtype in striatal cultures and indicate that these AC5 TM peptides are not able to destabilize the interactions between AC5 and the receptors but induce an alteration of the quaternary structure of the complex that impedes the simultaneous binding of $G\alpha s$ and $G\alpha i$ subunit to AC5, the canonical interaction. Thus, the correct intermolecular interaction between AC5 and the $A_{2A}R$–$D_2R$ heterotetramer is also a necessary condition for the presence of the canonical Gs–Gi interaction at AC5.

The ability of quinpirole to reduce cAMP accumulation induced by CGS21680 implies that Gi acts on a Gs-activated AC5. Thus, simultaneous binding of $Gs\alpha$ to the C2 domain and $Gi\alpha$ to the C1 domain of a single AC5 must occur[8]. This, in fact, agrees with the suggested complex of two $A_{2A}R$–$D_2R$ hetero-tetramers that simultaneously bind to the same AC5 molecule (see Fig. 4g). In this model AC5 acts as a link between two heterotetramers, which makes compatible the antagonistic canonical interaction between $Gs\alpha$ and $Gi\alpha$ activated proteins at the same AC5 molecule. Moreover, the membrane-spanning M1 and M2 domains of AC5 can accommodate between the two $A_{2A}R$–$D_2R$ heterotetramers (Supplementary Fig. 6), providing the frame for the series of experimentally determined TM contacts between $A_{2A}R$, $D_2R$, and AC5 (see above). However, with the model that includes two $A_{2A}R$–$D_2R$ heterotetramers and two AC5 molecules (Fig. 4g), only one AC5 simultaneously interacts with $Gs\alpha$ and $Gi\alpha$. This would imply that quinpirole could only produce a partial inhibition of CGS21680-induced cAMP accumulation, while the results showed in Fig. 5a–e demonstrate that quinpirole produces an almost complete blockade. We therefore propose that the minimal functional quaternary structure (see Fig. 4g) forms a linearly arranged high-order oligomeric structures (Supplementary Fig. 6e).

## Discussion

Striatal $A_{2A}R$ and $D_2R$ are known to form functionally and pharmacologically significant heteromers that modulate basal ganglia function[10]. Here, we demonstrate the existence of inter-molecular interactions between $A_{2A}R$, $D_2R$, and AC5 with the emergence of functional $A_{2A}R$–$D_2R$ heterotetramer-AC5 complexes. These complexes sustain the canonical Gs–Gi interaction at the AC level, the ability of a Gi-coupled GPCR to counteract AC activation mediated by a Gs-coupled GPCR.

We first identified the symmetrical TM 6 homodimer and TMs 4/5 heterodimer interfaces in the $A_{2A}R$–$D_2R$ heterotetramer from results of BiFC experiments obtained with specific TM peptides mimicking TM receptor domains. While BiFC complex formation under in vitro conditions has been considered to be essentially irreversible[24], several studies indicate that under in vivo conditions BiFC complex formation can be reversible[25–27]. The

present results provide additional support to that reversibility, which lies on the specificity of the peptide approach, demonstrated by the qualitative identical results from BiFC, PLA, and cAMP accumulation experiments. From these results, we could develop a computational model, where only the two internal protomers participate in the heteromeric interface and the two external pro-tomers participate in the homomeric interface of the $A_{2A}R$–$D_2R$ heterotetramer. A pattern of similar symmetrical interfaces of GPCR homomers and heteromers involving specific TMs has emerged from several studies also using TM interfering peptides, cross-linking techniques or crystallographic analysis (see ref.[6], for review). The consistent results we obtained with interfering pep-tides in experiments with biosensor-fused receptors in transfected cells and with native receptors in striatal neurons in culture, pro-vide strong support for the involvement of TM 6 in the homomeric interfaces and TM 4 and TM 5 in the heteromeric interface of the $A_{2A}R$–$D_2R$ heterotetramer in its natural environment. The differ-ences in the apparent interfaces of $A_{2A}R$ and $D_2R$ homomers here reported as compared to previous studies (TM 6 versus TMs 4 or/ and 5)[28,29], could be due to the different experimental approaches and, most likely, due to the presence of heteromeric partner receptors that influence the TM interfaces. The fact that rearran-gement of TM 6 constitutes main ligand-induced conformational changes that determine G protein activation and modulation of ligand affinity[30], provides a frame for the understanding of allos-teric communications through the protomers in GPCR oligomers[4–6]. Thus, in our constructed models, TM 6 in the inactive closed conformation of the unliganded protomer interacts with TM 6 in the active open conformation of the G protein-bound protomer (Supplementary Fig. 4).

An important conclusion from this study is that the non-activated or agonist-activated $A_{2A}R$–$D_2R$ heterotetramer is able to stablish different molecular interactions with AC5. By using specific interfering peptides, we demonstrate that these interac-tions involve TMs from the receptors and the AC5. The specifi-city of the peptide approach was unambiguously demonstrated with their orientation-dependent selectivity on their ability to destabilize the asymmetrical interfaces between AC5 and the receptors. The differential effect of interfering TM peptides in the absence and presence of agonists implies a major rearrangement of the membrane-spanning domains of the activated pre-coupled complex with an increase in the number of TMs of AC5 directly or indirectly involved in the oligomerization with the $A_{2A}R$–$D_2R$ heterotetramer during agonist exposure. This rearrangement could be driven by the agonist-induced relative movement of the $G\beta\gamma$ subunit away from the helical-domain of the $G\alpha$ subunit, simultaneously pulling the NT domain of AC5[16] and facilitating the interaction of its catalytic domains with the corresponding $G\alpha$ subunit[18,19]. This key role of the G protein in determining changes in the quaternary structure of the $A_{2A}R$–$D_2R$ heterotetramer-AC5 complex upon receptor activation would agree with the recently described stable pre-coupling of striatal Golf and AC5[22] and the here described less stable interactions between TMs of AC5 and $A_{2A}R$ and $D_2R$.

Probably the most significant conclusion of the study is that the $A_{2A}R$–$D_2R$ heterotetramer-AC5 complex sustains the canonical antagonistic Gs–Gi interaction at the AC level. This was also demonstrated with specific interfering TM peptides, by the very selective ability of the TM peptides that mimic the heteromeric interface in the $A_{2A}R$–$D_2R$ heterotetramer to block the canonical antagonistic interaction in striatal neurons in culture. The sig-nificant control of $A_{2A}R$ signaling by $D_2R$ implied that most $A_{2A}R$ that signal through AC5 are forming heteromers with $D_2R$ in this neuronal preparation. Previous studies indicate that the same situation occurs in vivo in the striatum, where the pharmacolo-gical or genetic blockade of $D_2R$ disinhibits adenosine-mediated

activation of AC in the striato-pallidal neuron[31]. In fact, $A_{2A}R$ blockade counteracts most biochemical and behavioral effects induced by interruption of $D_2R$ signaling[31]. In complete agreement are also the results obtained by Lee et al. with AC5 knockout mice[21], which show that AC5 is the principal AC integrating signals from $A_{2A}R$ and $D_2R$ in the striatum and that the signaling cascade involving AC5 is essential for the behavioral effects of $D_2R$ antagonists, and therefore antipsychotic drugs. The efficient $D_2R$-mediated antagonism of $A_{2A}R$-mediated AC activation, however, cannot be explained by a minimal functional structure of an $A_{2A}R$–$D_2R$ heterotetramer-AC5 complex that can sustain a canonical Gs–Gi interaction at AC, which is composed of two $A_{2A}R$–$D_2R$ heterotetramers and two AC5 molecules. Such a complex would not allow the $D_2R$ agonist to exert the almost complete inhibition of $A_{2A}R$ agonist-mediated cAMP revealed in the experiments on striatal neurons in culture. In fact, this quaternary structure suggests the possible formation of zig-zagged arranged high-order oligomeric structures (Supplementary Fig. 6), as proposed for other GPCRs, including $D_2R$ and rhodopsin homomers[20,32]. To our knowledge, these are the first data suggesting higher-order linear arrangements of GPCR heteromers and effectors.

The present study represents a proof of concept of the significant functional role of GPCR heteromers within a signalosome, since it demonstrates that GPCR heteromers provide the frame for biochemical interactions previously thought to be independent of intermolecular receptor–receptor interactions, on classical receptor cross-talk at the second-messenger level[33]. Therefore, we postulate that pre-coupling should not only apply to other Gs–Gi–AC-coupled heteromers, but also to heteromers coupled to other G proteins and effectors, such as the well-established Gi–Gq-coupled metabotropic glutamate receptor $mGlu_2$ receptor-serotonin 5-$HT_{2A}$ receptor heteromer[34], which could be pre-coupled to potassium channels[35]. At a more general level, the present results represent a very significant support to the still controversial concepts of GPCR pre-coupling and oligomerization.

## Methods

**Vectors and fusion proteins**. Sequences encoding amino acid residues 1–155 and 156–238 of YFP Venus protein were subcloned into the pcDNA3.1 vector to obtain the YFP Venus hemi-truncated proteins (pcDNA3.1-cVenus or pcDNA3.1-nVenus vectors). The cDNA constructs encoding human $A_{2A}R$ or $D_2R$ in pcDNA3 vectors were subcloned in pRluc-N1 (PerkinElmer, Wellesley, MA) to generate $A_{2A}R$-Rluc or $D_2R$-Rluc fusion proteins on the C-terminal end or were subcloned to be in-frame with restriction sites of pcDNA3.1-cVenus or pcDNA3.1-nVenus vectors to give the plasmids that express proteins fused to hemi-YFP Venus on the C-terminal end ($A_{2A}R$-cYFP, $D_2R$-cYFP, $A_{2A}R$-nYFP or $A_{2A}R$-nYFP). Human AC5 cDNA was amplified without its stop codon using sense and antisense primers harboring unique KpnI and EcoRV. The amplified fragment was subcloned to be in-frame with restriction sites of pEYFP-N1 (enhanced yellow variant of GFP; Clontech, Heidelberg, Germany) or pcDNA3.1-nVenus vectors to give the plasmids that express AC5 fused to YFP or hemi-YFP Venus on the C-terminal end (AC5-YFP or AC5-nYFP).

**Cell cultures and transfection**. Primary cultures of striatal neurons were obtained from fetal Sprague Dawley rats of 19 days. All experiments were carried out in accordance with EU directives (2010/63/EU and 86/609/CEE) and were approved by the Ethical Committee of the University of Barcelona. Striatal cells were isolated as described elsewhere[20] and plated at a confluence of 40,000 cells/0.32 cm². Cells were grown in Neurobasal medium supplemented with 2 mM L-glutamine, 100 U/ml penicillin/ streptomycin, and 2% (v/v) B27 supplement (GIBCO) in a 96-well plate for 12 days. HEK-293T cells were grown in Dulbecco's modified Eagle's medium (DMEM) supplemented with 2 mM L-glutamine, 100 U/ml penicillin/streptomycin, and 5% (v/v) heat inactivated fetal bovine serum (Invitrogen). HEK-293T cells (ATCC, Manassas, VA) were transfected with the plasmids encoding receptors by the PEI (PolyEthylenImine) method as previously described[20].

**TAT-TM peptides**. Peptides with the sequence of transmembrane domains (TM) of $A_{2A}R$ and $D_2R$ and putative TM peptides of AC5 fused to the HIV transactivator of transcription (TAT) peptide (YGRKKRRQRRR) were used as oligomer-destabilizing molecules. The cell-penetrating TAT peptide allows intracellular delivery of fused peptides[36]. The TAT-fused TM peptide can then be inserted effectively into the plasma membrane because of the penetration capacity of the TAT peptide and the hydrophobic property of the TM moiety[11]. To obtain the right orientation of the inserted peptide, the HIV-TAT peptide was fused to the C-terminus or to the N-terminus as indicated. The amino acid sequences of the fusion peptides are shown in Supplementary Table 2. Several algorithms were used to identify putative TMs in the primary amino acid sequence of AC5 (Supplementary Table 1).

**Bimolecular fluorescence complementation**. HEK-293T cells were transiently co-transfected with the cDNA encoding a protein fused to nYFP and a protein fused to cYFP. After 48 h, cells were treated or not with the indicated TM peptides (4 µM) for 4 h at 37 °C. The time of incubation and concentration of TM peptides were chosen from results of concentration-dependent and time-dependent response experiments of the possible BiFC destabilization by all seven TM peptides of the $A_{2A}R$ in HEK-293T cells transfected with $A_{2A}R$-nYFP and $A_{2A}R$-cYFP (Supplementary Fig. 3). The same parameters were applied with $D_2R$ and AC5 TM peptides and in PLA and cAMP experiments. To quantify protein reconstituted YFP Venus expression, cells (20 µg protein; 50,000 cells/well) were distributed in 96-well microplates (black plates with a transparent bottom, Porvair, King's Lynn, UK), and emission fluorescence at 530 nm was monitored in a FLUOstar Optima Fluorimeter (BMG Labtechnologies, Offenburg, Germany) equipped with a high-energy xenon flash lamp, using a 10-nm bandwidth excitation filter at 400 nm reading. Protein fluorescence expression was determined as the fluorescence of the sample minus the fluorescence of cells not expressing the fusion proteins (basal). Cells expressing protein-cVenus and nVenus or protein-nVenus and cVenus showed similar fluorescence levels than non-transfected cells.

**Bioluminescence resonance energy transfer assay**. HEK-293T cells were transiently cotransfected with a constant amount of expression vectors encoding for proteins fused to RLuc and with increasing amounts of the expression vectors corresponding to proteins fused to YFP. To quantify protein-YFP expression, cells (20 µg protein, around 50,000 cells/well) were distributed in 96-well microplates (black plates with a transparent bottom), and fluorescence was read in a Fluo Star Optima Fluorimeter (BMG Labtechnologies, Offenburg, Germany) equipped with a high-energy xenon flash lamp, using a 10-nm bandwidth excitation filter at 400 nm reading. Fluorescence expression was determined as fluorescence of the sample minus the fluorescence of cells only expressing the BRET donor. For BRET measurements, the equivalent of 20 µg of cell suspension was distributed into 96-well microplates (Corning 3600, white plates; Sigma) and 5 µM coelenterazine H (Molecular Probes, Eugene, OR) was added. The readings were taken 1 min later using a Mithras LB 940. The integration of the signals detected in the short-wavelength filter at 485 nm and the long-wavelength filter at 530 nm was recorded. To quantify protein-RLuc expression luminescence, readings were performed 10 min after adding 5 µM of coelenterazine H. Fluorescence and luminescence of each sample were measured before every experiment to confirm similar donor expressions (approximately 100,000 bioluminescence units) while monitoring the increase in acceptor expression (1000 to 30,000 fluorescence units). The net BRET is defined as [(long-wavelength emission)/(short-wavelength emission)] − Cf, where Cf corresponds to [(long-wavelength emission)/(short-wavelength emission)] for the donor construct expressed alone in the same experiment. BRET is expressed as milliBRET units (mBU; net BRET x 1000). Data were fitted to a nonlinear regression equation, assuming a single-phase saturation curve with GraphPad Prism software (San Diego, California, US). $BRET_{max}$ and $BRET_{50}$ values were obtained from the analysis of the BRET saturation curves. $BRET_{50}$ is a magnitude related to the affinity of the protein-protein interaction, with low values representing high affinity (as in the present results; Fig. 2a–d).

**Proximity ligation assay**. HEK293T cells or neuronal primary cultures were grown on glass coverslips and fixed in 4% paraformaldehyde for 15 min, washed with phosphate-buffered saline (PBS) containing 20 mM glycine, permeabilized with the same buffer containing 0.05% Triton X-100, and successively washed with TBS. Heteromers and AC5-receptor complexes were detected using the Duolink II in situ PLA detection Kit (OLink; Bioscience, Uppsala, Sweden) following supplier's instructions. A mixture of the primary antibodies [mouse or rabbit anti-$A_{2A}R$ antibodies (1:100; 05-717 and AB1559P, Millipore, Darmstadt, Germany), rabbit anti-$D_2R$ antibody (1:100; AB5084P, Millipore) and the recently characterized mouse anti-AC5 antibody[22] (1:50)] was used to detect $A_{2A}R$–$D_2R$ heteromers together with PLA probes detecting mouse or rabbit antibodies. The specificity of the same $A_{2A}R$ and $D_2R$ antibodies for PLA assays has been previously demonstrated[37]. Then, samples were processed for ligation and amplification with a Detection Reagent Red and were mounted using a DAPI-containing mounting medium. Samples were analyzed in a Leica SP2 confocal microscope (Leica Microsystems, Mannheim, Germany) equipped with an apochromatic 63X oil-immersion objective (1.4 numerical aperture), and 405-nm and 561-nm laser lines. For each field of view a stack of two channels (one per staining) and 4 to 6 Z-stacks with a step size of 1 µm were acquired. Images were opened and processed with Image J software (National Institutes of Health, Bethesda, MD). Quantification of the total number of red dots versus total cells (blue nuclei) was counted on the maximum projections of each image stack. After getting the projection, each channel was processed individually.

**Determination of cAMP**. Homogeneous time-resolved fluorescence energy transfer (HTRF) assays were performed using the Lance Ultra cAMP kit (Perki-nElmer), based on competitive displacement of a europium chelate-labeled cAMP tracer bound to a specific antibody conjugated to acceptor beads. We first established the optimal cell density for an appropriate fluorescent signal. This was done by measuring the TR-FRET signal as a function of forskolin concentration using different cell densities. The forskolin dose-response curves were related to the cAMP standard curve, to establish which cell density provides a response that covers most of the dynamic range of the cAMP standard curve. Cells (1000–2000 HEK-293T or 4000 to 5000 primary cultures per well) growing in medium containing 50 μM zardeverine were pre-treated with toxins or the corresponding vehicle in white ProxiPlate 384-well microplates (PerkinElmer) at 25 °C for the indicated time and stimulated with agonists for 15 min before adding 0.5 μM forskolin or vehicle and incubating for an additional 15 min period. Fluorescence at 665 nm was analyzed on a PHERAstar Flagship microplate reader equipped with an HTRF optical module (BMGLab technologies, Offenburg, Germany).

**Computational models**. Inactive models of the human $A_{2A}R$ and $D_2R$ were constructed based on the crystal structures of inactive $A_{2A}R$ (PDB id 5IU4)[38] and $D_3R$ (PDB id 3PBL)[39], respectively. The "active" conformations of $A_{2A}R$ bound to Gs and $D_2R$ bound to Gi were modeled by incorporating the active features of the crystal structure of $\beta_2$-adrenoceptor in complex with Gs (PDB code 3SN6)[30]. The globular α-helical domain of the α-subunit was modeled in the "closed" conformation, using the crystal structure of either Gsα (PDB id 1AZT)[40] or Giα (PDB id 3UMR)[41]. The absence of crystal structures of the M1 and M2 domains of AC or close protein templates impede their inclusion on the models. Nevertheless, the results with interfering peptides provide significant information about the putative location of the TM segments, which have been considered to form an antiparallel six-helix bundle with an elliptical ring shape as most of the membrane proteins. The structure of the intracellular C1 and C2 domains of AC in complex with Gsα and Giα was modeled as in the crystal structure of C1 and C2 in complex with Gsα (PDB id 1CUL)[42]. All homology models were built using Modeller 9.16[43]. The structure of $A_{2A}R$ and $D_2R$ heterodimer, using the TMs 4/5 interface, was modeled as in the oligomeric structure of the $\beta_1$-adrenoceptor (PDB code 4GPO)[44], whereas the structures for $A_{2A}R$ (inactive and Gs-bound "active" $A_{2A}R$) and $D_2R$ (inactive and Gi-bound "active" $D_2R$) homodimers were modeled using molecular dynamics simulations (see Supplementary Fig. 2) due to the absence of crystal structures of oligomers using exclusively the TM6 interface[11].

**Statistical information**. One-way ANOVA followed by Dunnett's or Tukey's multiple comparison tests were used for statistical comparisons between different groups of results. Number of experiments and replications as well as the statistical results are shown in the corresponding figure legends.

**Data availability**. All data that support the findings of this study are available from the corresponding author upon reasonable request.

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

## Acknowledgements

This study was supported by the Intramural funds of the National Institute Drug Abuse, grants from the Spanish "Ministerio de Economía y Competitividad" and European Regional Development Funds of the European Union (SAF2014-54840-R, SAF2015-74627-JIN, SAF2016-77830-R and SAF2017-84117-R), and "Fundació La Marató de TV3" (20140610) and Government of Catalonia Grant (2014-SGR-1236).

## Author contributions

G.N., Ar.C., V.C.-A., E.M. and N.-S.C. performed experiments. G.N., Ar.C., V.C.-A., E.M., N.-S.C., An.C. E.I.C, C.W.D, V.C., L.P., C.L. and S.F. performed data analysis and interpretation. G.N., Ar.C., C.W.D, V.C., L.P., C.L. and S.F. wrote and prepared the manuscript.

## Additional information

**Competing interests:** The authors declare no competing interests.

