## [Peer Review File · Nature Communications]

Reviewers' comments:

Reviewer #1 (Remarks to the Author):

This paper examines the nature of a stable complex consisting of a GPCR heterotetramer and AC5 equivalents. The authors use TAT-TM peptides mimicking various transmembrane domains of the A2A-adenosine receptor, the D2 dopamine receptor and AC5. They also use BiFC and PLA as primary outcome measures examining complex formation from different vantage points. Their premise is that GPCR homodimers form heterotetramers that contain effectors and presumably heterotrimeric G proteins in a stable, pre-assembled signalling complex. I am supportive of this idea but I have several concerns regarding the core premise and how they have performed and interpreted their data.

1) Using BiFC they "force" dimerization of either homodimer or the A2AR/D2R heterodimer. 48 hours after transfection they add TAT-TM peptides for 4 hours and test for their ability to disrupt complexes as measured by a loss of fluorescence. They may have confirmed that such tagged versions of their receptors and AC5 remain functional (which they should reference) or if not, they should demonstrate here. In my experience such reconstituted dimers do not come apart. So what do their experiments actually show? Prevention of the formation of new dimers? They should demonstrate this more directly. I am concerned that not all the TAT-TM peptides get into the membrane with the same efficacy- how do the authors control for that? The scrambled peptides that work do not give me confidence in this regard either.

2) If these forced dimers don't come apart, how do we interpret the effects of receptor ligands on fluorescence? BiFC is a crude tool to examine the conformational dynamics of protein complexes in response to ligands unless it can be demonstrated that such complexes can come together and apart in a dynamic equilibrium.

3) I have similar concerns for the PLA experiments- do they suggest the dimers are coming apart in response to TAT-TMs? This goes against the core argument the authors are trying to make. Also, the authors could demonstrate the PLA does not occur in the absence of receptor expression in HEK 293 cells.

4) I don't buy the notion that all heterotetramers are dimers of homodimers as the authors suggest in the introduction. The dynamics of GPCR complexes are highly variable depending on the molecular and cellular context assayed. Personally, I think metastability is the core feature of these complexes and I like the approach taken here to examine it. Perhaps the authors could be more

nuanced in the introduction. The question of proximity versus stability is the key issue here- perhaps a third technique like co-immunoprecipitation with and without TAT-TM or agonist could settle the issue?

5) The elephant in the room is what happens to G protein partners in these complexes? Are they important for formation of the R/R/E complexes or merely transducers of conformational information in response to agonist.

Minor comments

1) The word "strong" in the abstract is a relative term.

Reviewer #2 (Remarks to the Author):

The manuscript "Functional pre-coupled complexes of receptor heteromers and adenylyl cyclase" aims to address the existence of pre-coupling between GPCRs (A2A and D2R) and transmembrane adenylyl cyclase type 5 using engineered mimetic transmembrane domain peptides to disrupt interactions. Although it is quite accepted that GPCRs and G proteins must be in close proximity to their effectors this remains a fundamental question in the field not much is known on whether these different actors can form a complex in absence of receptor ligand. The study uses fluorescence-based technology to elucidate protein/protein organization in association with computational models and signaling experiments in both heterologous system and primary cells. Overall, this is an elegant work which provides new insights on how GPCR heteromers and AC5 may pre-form a functional complex and how ligand may rearrange these complexes. The authors should consider the following:

1- Authors assumed that TMs mimetic peptides are all correctly orientated and integrated in the membrane on the basis of the TAT-fused strategy. However, no evidences are provided. This is important to check especially for TM2,3 and 7 of A2A and D2R as they have no apparent effect on Bimolecular Fluorescence complementation.

2- How were determined the time of pretreatment (4 hours) and the concentration of peptide (4 responses may allow to establish the relative IC50 for each peptide as any other pharmacological tool.

3- BRET data suggest the formation of a complex between AC5 and D2R or AC5 and A2A. Although non-fluorescent receptor is added in the experiment to allow the formation of the heterotetramer, authors did not verify whether AC5 overexpression may affect the heteromerization of A2A with D2R. This needs to be verified.

4- In line with previous comment, AC5 TM peptides should be tested on A2A/D2R heteromer formation as negative controls as well as the effect of A2A TMs or D2R TMs on A2A/AC5 and D2R/AC5 respectively.

5- Authors proposed an “hypothetical” model in which AC5 and heterotetramers (A2A/D2R) associate in a linear manner (fig. 3g and supp fig. 3) suggesting a certain stoichiometry. This is an interesting hypothesis however, this is mostly based on overexpressed proteins. Considering the relative expression of both receptors in AC5 in striatal neurons, authors should comment on how this may also be true in native cells. PLA for AC5/D2R or AC5/A2A in neurons may reveal the pre-assemble complexes in a relevant cell model.

6- TM5n impaired association between receptors and AC5 is as good as TM5b, but not TM5 in absence of agonist. Additionally, authors hypothesized that TM5n may be part of intracellular IL2 of AC5 and involved in pre-coupling with receptors. Thus, a scramble IL2 peptide would be a good negative control here.

Minor comments:

1- Line 190. TM4n also behave as a negative control peptide and is not mentioned.

2- How the association between GPCRs and AC5 would change if agonists were added? Since TMs from M1 domain show a marked increased sensitivity to TM2,3,5 and 6 disrupting peptides, some changes in the association between AC5 and receptors may occur.

3- Would co-treatment with several peptides have an additive effect on A2A-D2R or AC5/receptors complexes?

Reviewer #3 (Remarks to the Author):

In this current study Navarro and co-workers addressed highly important questions in the GPCR field related to homo- or hetero-oligomerization and associated GPCR/effector-complex constitution. Higher order complexes between GPCR protomers, homo- or heteromeric, are in fact known and evidenced for many class A and other classes of GPCRs, but detailed information regarding structural prerequisites or functional consequences/relations of those complexes (GPCR interactome) is rarely available so far. Approaches to obtain experimental data deciphering molecular backgrounds

including structural (e.g. active versus inactive state conformations) and functional (signaling) aspects are still challenging with respect to appropriate methods.

To the authors credit they comprehensively studied focused issues, means a systematic “multiple-site” perspective on potentially contacting partners and functional consequences.

The techniques used in this study are appropriate, the received data are not over-interpreted in the conclusions and discussion. The context is described well throughout the manuscript and under consideration of previously and recently published information.

To the reviewers opinion this study can be seen as a progressive contribution and suggestion to several specific but also general aspects in the GPCR field. This concerns the TMH6-TMH6 interface between GPCR homomers, the simultaneous heterooligomeric dimer-dimer interface at TMH4-TMH5, the independence of homodimer-interfaces on the activity state, moreover the concluded pre-coupling or close spatial distance of AC to the inactive receptors, changes in receptor-AC interaction during activation, and finally implications for canonical signaling (Gs and Gi signaling in heteromeric GPCR complexes). The results may also have impact on parameters or mechanisms related to selectivity of GPCR signaling, but definitively they contribute information on pre-coupling partners of GPCRs, at least for the studied receptors.

Moreover, the manuscript is well balanced between the particular sections and the data are presented convincing.

Major points:

- Are data available proving on GPCR/AC interaction without available G-protein? Or is a G-protein/AC pre-complex a prerequisite for interaction with the GPCR (or a GPCR/G-protein pre-complex for interaction with AC, respectively)?

Minor points:

- The results and suggested scenario of receptor/AC pre-coupling must be strongly dependent (independently from the used method) also on the capacity of the investigated receptors to signal in a ligand-free (constitutive) basal state. It would be helpful to clarify this aspect for A2A and the D2R in the discussion section.
- It is of note, that both homodimers here are characterized by a TMH6-TMH6 interface. A discussion is needed how a TMH6-TMH6 interface in the homodimers may have impact on receptor activation (or how receptor activation can occur in such a constellation), because TMH6 outward movement is a significant feature of GPCR activation from a structural perspective and such a postulated interface may constrain this helix.
- Are there any indications that beside AC and here tested GPCRs other secondary effectors interact in pre-complexes with their respective GPCRs?

Answers to Reviewer #1

1. Using BiFC they "force" dimerization of either homodimer or the A2AR/D2R heterodimer. 48 hours after transfection they add TAT-TM peptides for 4 hours and test for their ability to disrupt complexes as measured by a loss of fluorescence. They may have confirmed that such tagged versions of their receptors and AC5 remain functional (which they should reference) or if not, they should demonstrate here. In my experience such reconstituted dimers do not come apart. So what do their experiments actually show? Prevention of the formation of new dimers? They should demonstrate this more directly. I am concerned that not all the TAT-TM peptides get into the membrane with the same efficacy- how do the authors control for that? The scrambled peptides that work do not give me confidence in this regard either.
2. If these forced dimers don't come apart, how do we interpret the effects of receptor ligands on fluorescence? BiFC is a crude tool to examine the conformational dynamics of protein complexes in response to ligands unless it can be demonstrated that such complexes can come together and apart in a dynamic equilibrium.

In these two points, the reviewer expresses his/her concerns about two assumptions regarding BiFC. First, that fusion of the two complementary YFP fragments "forces" the intermolecular interaction between the proteins to which they are separately attached. In fact, it is the complex formation between the two interacting proteins what forces the fusion of the YFP fragments (Rose et al., 2010, Brit J Pharmacol, 159, 738). This is usually demonstrated by control experiments showing the lack of fusion of non-attached complementary YFP fragments or separately attached to non-interacting proteins (as similar as possible to the ones showing positive BiFC). For instance, we previously showed with BiFC that the positive interacting A2AR and D2R do not interact with D1R and A1R, respectively (Bonaventura et al., 2015, cited in the text).

The second assumption about BiFC is that once the YFP fragments are fused, BiFC is irreversible. While BiFC complex formation under in vitro conditions has been considered to be essentially irreversible (Rose et al., 2010), several studies indicate that under in vivo conditions BiFC complex formation can be reversible. This has for instance been demonstrated by Schmidt et al. (2003, Mol Cell 12, 1287), in their study about the differential association and dissociation of specific complexes of NF- κ B with two different isoforms of the I- κ B inhibitor; also by Guo et al. (2005, J Biol Chem, 280, 1438), in experiments about the $\beta\gamma$ -dependent dissociation of complexes of phospholipases C β_2 and C δ_2 ; and by Anderie and Schmid (2007, Cell Biol Int, 31, 1131), with visualizing in vivo of actin cytoskeleton dynamics (reversible actin/actin BiFC complexes). But also, our results constitute a significant demonstration of reversible BiFC. And this lies on the specificity of the peptide approach, demonstrated by parallel experiments of BiFC and PLA. However, the reviewer also expresses concern about a possible differential efficacy of the peptides that could depend on a differential efficacy to integrate in the plasma membrane. We agree with the reviewer about the possibility that this integration can be

different among peptides, as we particularly discuss in the manuscript when evaluating hydrophathy values in order to explain the positive effect of the two differently oriented peptides mimicking putative TM 5 of AC5 (Table 1). Nevertheless, we should stress that all the other putative and non-putative TM peptides are very similar in their chemical characteristics. Furthermore, we are not evaluating quantitative effects (degree of disruption), but qualitative effects (presence or absence of disruption) of the peptides. Then, we strongly believe that our qualitative results demonstrate the selectivity of the peptide approach. For instance, from the 14 different TM peptides of A2AR and D2R, with very similar chemical characteristics, only one peptide, TM6 of A2AR (but not of D2R), disrupts A2AR-A2AR dimerization, and only one peptide, TM6 of D2R (but not A2AR), disrupts D2R-D2R dimerization. Then, four different TM peptides, TM4 and TM5 of A2AR and D2R, selectively disrupt A2AR-D2R heteromerization, and those selective peptides are the only ones able to disrupt A2AR-D2R complexes in striatal cells in culture by using PLA. Importantly, this specificity provided the information for only one solution in the computational modeling of the heterotetramer and, even more sound, the very same peptides that disrupted A2AR-D2R heteromerization (as demonstrated by BiFC and PLA) were the only ones able to disrupt the functional canonical interaction at the AC5 level. We believe that the validity of the peptide approach is also significantly supported (if not unambiguously demonstrated) with the analysis of all possible putative TMs of AC5, because of their orientation-dependent selectivity on their ability to destabilize oligomerization of AC5 with A_{2A}R and D₂R (in BiFC experiments), which allowed not only predicting asymmetrical interfaces between TMs of the receptors and AC5, but also providing a new putative topology of the TMs of the M1 domain of AC5. The fact is that the use of synthetic TM peptides is becoming a successful emergent tool to analyze the interface of GPCR oligomers (not only from our research group; see, for instance, Jastrzebska et al., J Biol Chem, 2015, 290, 25728; cited in the text).

These arguments (and references) that stress the ability of TM peptides to specifically disrupt BiFC and PLA in mammalian cells will now be included in a few sentences in the Discussion. Following the reviewer's advice, we have added control experiments (cAMP accumulation) demonstrating the functionality of all receptors attached to the full or fragmented YFP (Supplementary Fig. 1). In the Results section, we have also added a sentence and reference about the previously demonstrated normal functionality of AC5-YFP (Sadana et al., 2009).

- 3. I have similar concerns for the PLA experiments- do they suggest the dimers are coming apart in response to TAT-TMs? This goes against the core argument the authors are trying to make. Also, the authors could demonstrate the PLA does not occur in the absence of receptor expression in HEK 293 cells**

As mentioned above (and now made more explicit in the text), the specificity of the peptide approach is demonstrated by the identical qualitative results shown by BiFC and PLA experiments, which validate the ability of TM peptides to disrupt GPCR oligomeric interfaces. Thus, it is even more improbable that PLA detects "forced" complexes that

*become irreversible with the experimental procedure, as suggested for BiFC, due to the fact that non-fused proteins are detected in PLA experiments. Also important was the demonstration of the same PLA results with transfected receptors in HEK cells and in striatal cells in culture. **In the Supplementary Information (Supplementary Fig. 3a and 3b), we have also included the controls indicated by the reviewer, showing the lack of A2AR-D2R complexes in HEK cells only expressing A2AR or D2R.***

- 4. I don't buy the notion that all heterotetramers are dimers of homodimers as the authors suggest in the introduction. The dynamics of GPCR complexes are highly variable depending on the molecular and cellular context assayed. Personally, I think metastability is the core feature of these complexes and I like the approach taken here to examine it. Perhaps the authors could be more nuanced in the introduction. The question of proximity versus stability is the key issue here- perhaps a third technique like co-immunoprecipitation with and without TAT-TM or agonist could settle the issue?**

*We agree with the reviewer about being less categorical in the introduction about the evidence for heterotetramers as composed of heteromers of homodimers. Therefore, in the Introduction, **we replaced “indicate” by “suggest”**, when telling about growing evidence for GPCR homodimers as a common GPCR unit. And, in the sentence “heteromers can be viewed as constituted by different interacting homodimers”, **we replaced “can” by “could”**. About the questions of proximity and stability of putative protein-protein interaction, we totally agree with the reviewer about the need for more than one technique. That is why we used BiFC and PLA for the interactions between GPCRs and BiFC and BRET for the interactions between GPCRs and AC5, which in our experience and that of other research groups, there are more valuable methods to study direct interactions between GPCRs or GPCRs and signaling proteins than co-IP. The reviewer would probably agree with the fact that co-IP experiments with and without peptides would not lead to unequivocal information. Indirect interactions cannot be ruled out with co-IP, making almost impossible to know whether all the proteins found form one single complex with the bait protein, or whether they exist in different sub-complexes. Nevertheless, **a third technique, BRET plus double complementation of BRET biosensors, had already been used to study A2AR-D2R heterotetramers in a previous study referenced in the text (Bonaventura et al., 2015). Also, and in agreement with reviewer #2, a third technique, PLA, was now used to demonstrate complexing of endogenous AC5 with A2AR and with D2R in striatal cells (Supplementary Fig. 7).***

- 5. The elephant in the room is what happens to G protein partners in these complexes? Are they important for formation of the R/R/E complexes or merely transducers of conformational information in response to agonist.**

This is an important question already addressed in the first version of the manuscript, that obviously needs to be improved. In fact, the same point was also raised by reviewer

#2 (2nd minor comment) and #3 (major point). **The following sentences have been added or modified in the Results and Discussion sections (and a new reference included):** “The results indicate TM peptides that disrupt heteromerization do not disrupt the individual functional interactions between the receptors and AC5 in striatal cells in culture, most probably because of stable pre-coupling between the G proteins and AC5, as recently demonstrated for the specific $G\alpha_{olf}\beta_2\gamma_7$ -AC5 complex in the rodent striatum (Xie et al., 2015, *Elife* 4, 10451)”. “This key role of the G protein in determining changes in the quaternary structure of the $A_{2A}R$ - D_2R heterotetramer-AC5 complex upon receptor activation would agree with the recently described stable pre-coupling of striatal Golf and AC5 and the here described less stable interactions between TMs of AC5 and $A_{2A}R$ and D_2R ”. As already included in the first version of the manuscript, those changes can then be explained by the agonist-induced movement of the $G\beta\gamma$ subunit away from the helical-domain of the $G\alpha$ subunit, simultaneously pulling from the NT domain of AC5 and facilitating the interaction of its catalytic domains with the corresponding $G\alpha$ subunit.

6. Minor comment: The word "strong" in the abstract is a relative term.

We deleted “strong” from the abstract

Answers to Reviewer #2

- 1. Authors assumed that TMs mimetic peptides are all correctly orientated and integrated in the membrane on the basis of the TAT-fused strategy. However, no evidences are provided. This is important to check especially for TM2,3 and 7 of A_{2A} and D_2R as they have no apparent effect on Bimolecular Fluorescence complementation.**

*The ability of TAT-TMs to integrate in the plasma membrane in an orientation-specific manner was previously demonstrated by the elegant studies of He et al. (2011, *Neuron*, 69, 120; previously cited in the Methods section and **now explicitly mentioned in the Results section**), when determining the interface of the mu-delta-opioid receptor heteromer. Their method implied the immuno-cytochemical detection of TM peptides with GST fused to the N-terminus and TAT fused to either the N- or the C-terminus in permeabilized and non-permeabilized neuronal preparations. The localization of the TAT sequence determined the localization of GST, facing the intra- or the extracellular space. The corresponding TAT-TM peptides were always integrated in the membrane, but when facing the cytosol, GST could only be visualized in the permeabilized preparation. Therefore, we believe it should not be necessary to replicate these findings, which should require a large number of experiments. In addition, as mentioned in the response to point 1 by reviewer #1 (**and also now mentioned in the Results section**), we strongly believe that our qualitative results demonstrate the selectivity of the peptide approach. Indeed, the specificity of these results provided the information for only one solution in the computational modeling of the heterotetramer and the very same peptides that*

disrupted A2AR-D2R heteromerization (as demonstrated by two methods, BiFC and PLA) were the only ones able to disrupt the functional canonical interaction at the AC5 level. Additional positive results with the negative TM peptides (TM2, 3 and TM7) would have probably yielded no computational solution.

- 2. How were determined the time of pretreatment (4 hours) and the concentration of peptide (4 μ M)? Moreover, Dose-responses may allow to establish the relative IC50 for each peptide as any other pharmacological tool.**

*About the justification for the optimal concentration and time of incubation of the TM peptides, we had in fact performed concentration- and time-dependent response experiments of the possible BiFC disruption by all seven TM peptides of the A2AR in cells transfected with A2AR-nYFP and A2AR-cYFP. **The results are now included in Supplementary Fig. 2, and show that the selective disruptive effect of TM6 of A2AR is optimal at the concentration of 4 μ M and with a pretreatment of 4 hours.** In view of their very similar physicochemical properties, we considered that the same parameters could be used for TM peptides of the D2R and AC5. In fact, we obtained differential specific positive effects with those peptides in BiFC experiments which allowed totally congruent results with PLA and cAMP experiments.*

- 3. BRET data suggest the formation of a complex between AC5 and D2R or AC5 and A2A. Although non-fluorescent receptor is added in the experiment to allow the formation of the heterotetramer, authors did not verify whether AC5 overexpression may affect the heteromerization of A2A with D2R. This needs to be verified.**

*Following the reviewer's comment, as an additional control, we also verified if overexpression of AC5 could alter A2AR-D2R heteromerization with BRET experiments in HEK-293T cells transfected with A2AR-Rluc (0.3 μ g) and D2R-YFP (0.3 μ g) and increasing amounts of AC5 cDNA. No BRET differences were observed between the results obtained with 0, 0.3, 1.0 and 3.0 μ g of AC5 cDNA (56 ± 7 , 53 ± 6 , 53 ± 3 and 52 ± 4 mBU, respectively). **These additional data have been added in the Results section.***

- 4. In line with previous comment, AC5 TM peptides should be tested on A2A/D2R heteromer formation as negative controls as well as the effect of A2A TMs or D2R TMs on A2A/AC5 and D2R/AC5 respectively.**

*Following the reviewer's comment, as additional negative controls, we also tested AC5 TM1 to TM12 peptides on A2AR-nYFP-D2R-cYFP complementation and all the D2R TM and A2AR TM peptides on AC5-nYFP-A2AR-cYFP and AC5-nYFP-D2R-cYFP complementation, respectively, in the absence of ligands (**Supplementary Fig. 5**). **Sentence included in the Results Section.***

- 5. Authors proposed an "hypothetical" model in which AC5 and heterotetramers (A2A/D2R) associate in a linear manner (fig. 3g and supp fig. 3) suggesting a certain**

stoichiometry. This is an interesting hypothesis however, this is mostly based on overexpressed proteins. Considering the relative expression of both receptors in AC5 in striatal neurons, authors should comment on how this may also be true in native cells. PLA for AC5/D2R or AC5/A2A in neurons may reveal the pre-assemble complexes in a relevant cell model.

Following the reviewer's advice, PLA experiments were also performed with a recently characterized AC5 antibody (Xie et al., 2015). A_{2A}R-AC5 and D₂R-AC5 complexes could be revealed as red punctate staining in neuronal cells (Supplementary Fig. 7). This information and the corresponding additional methods have been added in the text.

6. **TM5n impaired association between receptors and AC5 is as good as TM5b, but not TM5 in absence of agonist. Additionally, authors hypothesized that TM5n may be part of intracellular IL2 of AC5 and involved in pre-coupling with receptors. Thus, a scramble IL2 peptide would be a good negative control here.**

Following the reviewer's advice, a scrambled TM5-TM5n peptide was synthesized (AC5-TM5s in Supplementary Table 1). As expected, it did not decrease AC5-nYFP-A_{2A}R-cYFP or AC5-nYFP-D₂R-cYFP complementation in the absence of ligands. This information is now included in the Results Section.

7. **Minor comments:**

1- Line 190. TM4n also behave as a negative control peptide and is not mentioned.

The following sentence has been added in the text: "The same as TM4, TM4n did not have a significant effect, and TM2n, TM3n and TM6n did behave as negative controls to their opposite-oriented peptides, since they did not decrease AC5-nYFP-A_{2A}R-cYFP or AC5-nYFP-D₂R-cYFP complementation..."

2- **How the association between GPCRs and AC5 would change if agonists were added? Since TMs from M1 domain show a marked increased sensitivity to TM2,3,5 and 6 disrupting peptides, some changes in the association between AC5 and receptors may occur.**

We have extended the information previously included in the sentences from lines 161-164. In addition to the respective loss of effect of TM1 of A_{2A}R and D₂R with CGS21680 and quinpirole, we have added that, apart from TM5 and TM6 of both A_{2A}R and D₂R, TM7 of A_{2A}R and TM2 of D₂R decreased fluorescence complementation in the presence of CGS21680 and quinpirole, respectively.

3- **Would co-treatment with several peptides have an additive effect on A_{2A}-D₂R or AC5/receptors complexes?**

As also mentioned in the answer to the first point of reviewer #1, our approach with the analysis of multiple TM peptides was designed to evaluate their individual qualitative effects (presence or absence of disruption), to determine the interface of A2AR-D2R heterotetramers and their interface with AC5. We agree with the reviewer that it would be of interest to study combinations of TM peptides, but that would require a large number of experiments that we believe is out of the scope of the manuscript.

Answers to Reviewer #3

1. Are data available proving on GPCR/AC interaction without available G-protein? Or is a G-protein/AC pre-complex a prerequisite for interaction with the GPCR (or a GPCR/G-protein pre-complex for interaction with AC, respectively)?

Please, see answer to the same point raised by reviewer #1 (point 5).

2. Minor points:
 - The results and suggested scenario of receptor/AC pre-coupling must be strongly dependent (independently from the used method) also on the capacity of the investigated receptors to signal in a ligand-free (constitutive) basal state. It would be helpful to clarify this aspect for A2A and the D2R in the discussion section.

*This is an important point we are in fact investigating. We have obtained experimental results that indicate that the previously described constitutive activity of the A2AR (Fernando-Duenas et al., ACS Chem Biol, 2014, 9, 2496) is absent in the A2AR-D2R heteromer. We would obviously want to keep this information for our manuscript in preparation. Nevertheless, **we have added the following short sentence in the Discussion**: “It can also be predicted that the constrains imposed by oligomerization should alter the constitutive activity of A2AR and D2R”.*

- It is of note, that both homodimers here are characterized by a TMH6-TMH6 interface. A discussion is needed how a TMH6-TMH6 interface in the homodimers may have impact on receptor activation (or how receptor activation can occur in such a constellation), because TMH6 outward movement is a significant feature of GPCR activation from a structural perspective and such a postulated interface may constrain this helix.

*We would like to thank the reviewer for raising this key issue that is now **addressed in the Discussion and Supplementary Figure 4**. As noted by the reviewer, agonists increase signaling by opening an intracellular cavity, required for the binding of the C-terminal $\alpha 5$ helix of the G protein, through the outward movement of TMs 5 and 6. Because one protomer can allosterically modulate the functional properties of the interacting receptor (cross-antagonism and negative cross-talk) (see Bonaventura et al., PNAS, 2015, 112, E3609; cited in the text), these TMs involved in the homo or*

heterodimerization interfaces provide a frame for the understanding of allosteric communications through the protomers in GPCR oligomers. For instance, we have suggested that formation of a very stable four-helix bundle via the TMs 5/6 interface, as observed in the crystal structure of the μ -opioid receptor, causes cross-antagonism (Viñals et al., PLoS Biol, 2015, 13, e1002194), because agonists cannot surmount this very stable, high surface complementarity dimer interface, and trigger the outward movement of TMs 5 and 6 for receptor activation. In contrast, in the constructed models of the present work, TM 6 in the inactive closed conformation of the unliganded protomer interacts with TM 6 in the active open conformation of the agonist-bound protomer (Supplementary Figure 4). The reviewer is totally right when noting that simultaneous outward movements of TM 6 in the homodimer is not feasible due to a steric clash between active open conformation of both TM 6. Likewise, simultaneous binding of two G proteins to the homodimer would not be possible due to a steric clash between both bulky G proteins. Thus, in the models of the A_{2A}R and D₂R homodimers only one protomer (active open TM 6 conformation) binds both the agonist and the α 5 helix of the G-protein.

- Are there any indications that beside AC and here tested GPCRs other secondary effectors interact in pre-complexes with their respective GPCRs?

*We are grateful for this comment of the reviewer, which allow us to generalize our findings to other putative pre-coupled complexes of GPCR, G proteins and effectors. **We have therefore expanded the end of the Discussion with the following sentence (and additional references):** “Therefore, we postulate that pre-coupling should not only apply to other Gs-Gi-AC-coupled heteromers, but also to heteromers coupled to other G proteins and effectors, such as the well-established Gi-Gq-coupled metabotropic glutamate receptor mGlu₂ receptor-serotonin 5-HT_{2A} receptor heteromer, which could be pre-coupled to potassium channels.*

REVIEWERS' COMMENTS:

Reviewer #1 (Remarks to the Author):

The authors have provided a thoughtful response to critiques raised. They argue strongly the BiFC is reversible but don't demonstrate this directly. I acknowledge other studies that show such peptide approaches work I just remained unconvinced that the authors show that their peptides can break up pre-existing complexes, rather than prevent the formation of new ones.

Reviewer #2 (Remarks to the Author):

Authors provided most of the additional controls required answered my questions.

In my opinion, the current revised version of the paper appears suitable for publication.

Answer to reviewer 1:

The authors have provided a thoughtful response to critiques raised. They argue strongly the BiFC is reversible but don't demonstrate this directly. I acknowledge other studies that show such peptide approaches work I just remained unconvinced that the authors show that their peptides can break up pre-existing complexes, rather than prevent the formation of new ones.

We agree with the reviewer and realize that this is mostly a semantic problem. In fact, we also believe that TM peptides do not break up pre-existing complexes. Taking into account the membrane fluidity and the motion characteristics of intrinsic membrane proteins, including GPCRs (see our recent publication from Navarro et al. BMC Biology, 2016, 14, 26) we do not believe on static receptor complexes. The same as several other researches in the field, we believe in a dynamic association and dissociation of the different elements in the complex. Molecular dynamic studies from Marta Filizzola's group (see, for instance, Provasi et al., PLOS Comp Biol, 2015, 11, e1004148) and single-molecule tracking studies from Martin Lohse's group (see, for instance, Calebiro et al., PNAS, 2013, 110, 743) strongly suggest that GPCR oligomers tend to associate and dissociate within a second-scale rate (also addressed in our review in Parmacol Rev). It is therefore totally conceivable that, within a multimolecular complex, formed by homodimers, heteromers, G proteins and effectors (to say the least), its elements constantly associate and re-associate within the same complex (since they establish intermolecular interactions with at least two different elements of the complex). Within this recognized theoretical framework, TM peptides, upon dissociation of the specific elements in the complex, would be able to establish an intermolecular interaction with the corresponding specific protein in the complex competing with the native protein and would then **interfere** with a re-association, **destabilizing** the quaternary structure of the complex. We would agree that this scenario needs to be substantiated experimentally, but this will require from extensive experimental and computational analysis, which we are in fact planning to address in the near future. Nevertheless, we realize that we should change some terms, like "disrupting", which seems to imply that peptides can break-up complexes. We have therefore changed the term "disrupting peptide" to "**interfering peptide**", and we have changed "disrupting effect" to "**destabilizing effect**" of the quaternary structure of the complex (of the heterotetramer and heterotetramer-AC5 oligomer). Finally, we also changed the term "disrupting effect" for "**blocking effect**", when expressing the effect of the peptide on the heterotetramer-AC5 function, on the canonical interaction.